# HARD-META-DATASET++: Towards Understanding Few-Shot Performance on Difficult Tasks

**Samyadeep Basu** *, **Megan Stanley, John Bronskill, Soheil Feizi, Daniela Massiceti**
{sbasu12, sfeizi}@umd.edu, {jfb54}@cam.ac.uk
{meganstanley, dmassiceti}@microsoft.com

## ABSTRACT

Few-shot classification is the ability to adapt to any new classification task from only a few training examples. The performance of current top-performing few-shot classifiers varies widely across different tasks where they often fail on a subset of 'difficult' tasks. This phenomenon has real-world consequences for deployed few-shot systems where safety and reliability are paramount, yet little has been done to understand these failure cases. In this paper, we study these difficult tasks to gain a more nuanced understanding of the limitations of current methods. To this end, we develop a general and computationally efficient algorithm called FASTDIFFSEL to extract difficult tasks from any large-scale vision dataset. Notably, our algorithm can extract tasks at least 20x faster than existing methods enabling its use on large-scale datasets. We use FASTDIFFSEL to extract difficult tasks from META-DATASET, a widely-used few-shot classification benchmark, and other challenging large-scale vision datasets including ORBIT, CURE-OR and OBJECTNET. These tasks are curated into HARD-META-DATASET++, a new few-shot testing benchmark to promote the development of methods that are robust to even the most difficult tasks. We use HARD-META-DATASET++ to stress-test an extensive suite of few-shot classification methods and show that state-of-the-art approaches fail catastrophically on difficult tasks. We believe that our extraction algorithm FASTDIFFSEL and HARD-META-DATASET++ will aid researchers in further understanding failure modes of few-shot classification models.

## 1 INTRODUCTION

Few-shot classification is the ability to distinguish between a set of novel classes when given only a few labelled training examples of each class (Lake et al., 2011; Fei-Fei et al., 2006). This holds potential across many real-world applications – from robots that can identify new objects (Ren et al., 2020), to drug discovery pipelines that can predict the properties of new molecules (Stanley et al., 2021). A few-shot *image* classifier is given a few labelled training images of the new object classes, called the support set. Once the classifier has adapted to this support set, it is then evaluated on novel test images of those classes, called the query set. Together, the support and query set is called a task.

Recent years have seen rapid progress in few-shot image classification (Snell et al., 2017; Finn et al., 2017b; Ye et al., 2020; Li et al., 2021; Radford et al., 2021; Kolesnikov et al., 2019), however, current top-performing methods display a wide range in performance over different tasks at test time (Fu et al., 2022; Agarwal et al., 2021). On META-DATASET (MD), a widely-used few-shot classification benchmark (Triantafillou et al., 2019), state-of-the-art classifiers obtain accuracies as low as 22% on some individual tasks though their average task accuracy is >55% (see Fig 1). Few works have undertaken a detailed examination of these 'difficult' tasks, yet they remain critical to interrogate for both future algorithmic development and the safety and reliability of deployed systems.

This paper aims to gain a more nuanced understanding of these 'difficult' tasks and the limitations of current methods. We define a difficult task as one on which a few-shot classifier performs poorly on the task's query set, after being adapted to its support set. Current methods for finding supports sets that lead to poor query performance rely on greedy search-based algorithms (Agarwal et al., 2021). These approaches, however, incur a high computational cost when sampling for a large numbers of

---

*Work done partly during a research internship at Microsoft Research, Cambridge (UK)

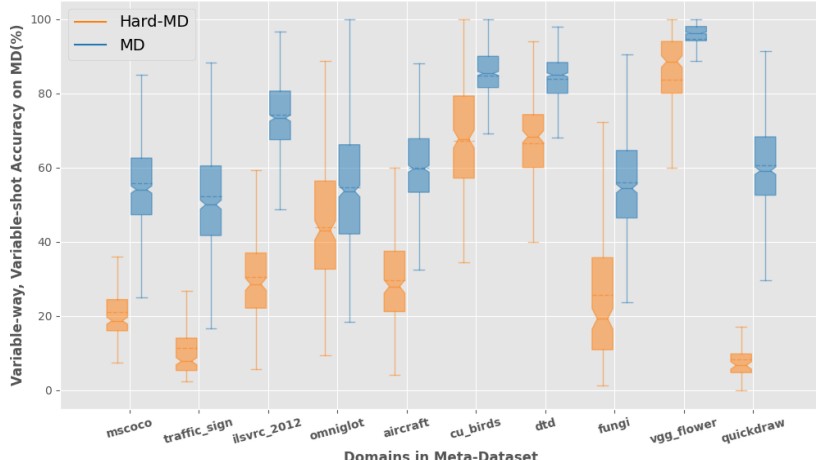

Figure 1: **A state-of-the-art method (Hu et al., 2022) performs consistently worse on difficult tasks in the MD split (HARD-MD) of HARD-META-DATASET++ compared to tasks in META-DATASET (MD) across all 10 MD sub-datasets.** The method uses ViT-S initialized with self-supervised DINO weights and is further meta-trained with ProtoNets on MD's ilsvrc_2012 split.

tasks, and for tasks with large support sets, as are common (and best) practices in few-shot evaluation protocols. As a result, the study of difficult tasks has been limited to small-scale datasets which lacks the challenging examples and the setup of large benchmarks such as META-DATASET.

To address this, we develop a general and computationally efficient algorithm called FASTDIFFSEL to extract difficult tasks from any large-scale dataset. Given a (meta-)trained few-shot classifier, a query set, and a search pool of support images, we formulate a constrained combinatorial optimization problem which learns a selection weight for each image in the pool such that the loss on the query set is maximised. The top-$k$ (i.e. most difficult) images per class are then extracted into a support set and paired with the query set to form a difficult task. This optimization can be repeated to obtain any number of difficult tasks. In practice, we find that FASTDIFFSEL is at least 20x faster than existing greedy search-based algorithms (Agarwal et al., 2021), with greater gains as the support pools and support set sizes increase.

We leverage the scalability of FASTDIFFSEL to extract difficult tasks from a wide range of large-scale vision datasets including META-DATASET (Triantafillou et al., 2019), OBJECTNET (Barbu et al., 2019), CURE-OR (Temel et al., 2018) and ORBIT (Massiceti et al., 2021) and collect these tasks into a new testing set called HARD-META-DATASET++ (HARD-MD++). The addition of datasets beyond META-DATASET is motivated by their real-world nature and that they provide image annotations of quality variations (e.g. object occluded, blurred, poorly framed) to enable future research into *why* a task is difficult. We provide early insights into this question in our analyses.

We stress test an extensive suite of state-of-the-art few-shot classification methods on HARD-MD++, cross-validating the difficulty of our extracted tasks across these top-performing methods. In Fig 1, we show one such method (Hu et al., 2022) performing consistently worse on the META-DATASET test split in HARD-MD++ than on the original MD test split across all 10 sub-datasets. In Section 5, we find that this trend holds true across a wide-range of few-shot classification methods.

We release HARD-MD++ along with a broad set of baselines to drive future research in methods that are robust to even the most difficult tasks. In summary, our contributions are the following:

1. FASTDIFFSEL, an efficient algorithm to extract difficult tasks from any large-scale vision dataset.
2. HARD-META-DATASET++, a new test-only few-shot classification benchmark composed of difficult tasks extracted from the widely-used few-shot classification benchmark META-DATASET and other large-scale real-world datasets: OBJECTNET, CURE-OR and ORBIT.
3. Extensive stress testing and novel empirical insights for a wide range of few-shot classification methods, including transfer- and meta-learning based approaches on HARD-MD++.

## 2 FEW-SHOT CLASSIFICATION: PRELIMINARIES AND NOTATIONS

A few-shot classification task is typically composed of (i) a support set $\mathcal{S}$ which contains a few labelled examples from a set of $N$ classes (e.g., $k_j$ examples for each class index $j \in [1, N]$)

---

**Algorithm 1** FASTDIFFSEL: Efficient algorithm for extracting a difficult few-shot task

---

**Require:** $\mathcal{Q}$ : task query set; $N$: number of classes (way), $\mathcal{P}$ : search pool for extracting task support set; $f_\theta$: (meta-)trained base model; $M$: size of search pool; $\alpha$: learning rate; $\{k_j\}_{j=1}^N$: set containing number of shots per class.

$\mathbf{w} \leftarrow \text{CONCAT}(\mathbf{w}_j) \qquad \forall j \in [1, N]$     ▷ Concatenate randomly initialized vectors for each class

**for** $j$ in $N$ **do**

    $c_j \leftarrow \sum_{i=1}^{|\mathcal{P}_j|} w_j^i f_\theta(x_j) / \sum_{i=1}^{|\mathcal{P}_j|} w_j^i$     ▷ Compute weighted class prototypes for each class

**end for**

$c \leftarrow [c_1, ..., c_N]$     ▷ Store the weighted prototypes for each class

$L \leftarrow \text{PROTO-LOSS}(\mathcal{Q}, c, f_\theta)$     ▷ Compute prototypical loss (Snell et al., 2017)

$L.\text{BACKWARD}(\mathbf{w})$     ▷ Compute gradients with respect to selection weights

$\mathbf{w} \leftarrow \mathbf{w} + \alpha \nabla_\mathbf{w} L(\mathbf{w})$     ▷ Gradient ascent for updating weights

$\mathbf{w}_j \leftarrow \text{PROJ}(\mathbf{w}_j, k_j) \qquad \forall j \in [1, N]$     ▷ Projection step per class

$s_j \leftarrow \text{EXTRACT}(k_j, \mathbf{w}_j, \mathcal{P}) \quad \forall j \in [1, N]$     ▷ Extract $k_j$ examples with the highest weights

$\mathcal{S} \leftarrow \text{CONCAT}(s_j) \qquad \forall j \in [1, N]$     ▷ Obtain the final difficult support set $S$

---

and (ii) a query set $\mathcal{Q}$ which contains a disjoint set of test examples for each of those $N$ classes (e.g., $q_j$ examples for each class $j \in [1, N]$). Given a trained base model $f_\theta$, the goal in few-shot classification is to use $\mathcal{S}$ to adapt $f_\theta$ to the task such that the model can then make predictions on the unseen examples in $\mathcal{Q}$. This is typically done for many tasks which are randomly sampled from a given dataset $\mathcal{D} = \{(x_i, y_i)\}_{i=1}^T$ consisting of $T$ examples and $J$ classes in total, where $x$ is the input image and $y$ is the corresponding class label. In few-shot sampling, tasks are constructed by first sampling a set of $N$ classes from the larger set of $J$ classes and subsequently sampling the support and query set. Tasks are typically referred to as $N$-way, $k$-shot tasks if $k_j = k, \forall j$. However, if the number of classes $N$ varies across tasks and the number of shots per class varies across the $N$ classes ($k_j, \forall j \in [1, N]$), they are referred to as variable-way, variable-shot tasks. Usually the number of query examples is kept fixed across all the classes (i.e. $q_j = q, \forall j \in [1, N]$).

**Base model training**. The underlying model $f_\theta$ is typically trained using one of two approaches: (i) meta-learning (Snell et al., 2017; Lee et al., 2019; Finn et al., 2017a) which involves training the model in an episodic manner on tasks sampled from a base training dataset, or (ii) transfer learning (Tian et al., 2020; Hu et al., 2022; Chen et al., 2019) which involves first pre-training a feature extractor on a large base dataset in an supervised or self-supervised manner, and then fine-tuning the final classification layer (or the entire model) at test time for each new test task.

## 3 FASTDIFFSEL: AN EFFICIENT ALGORITHM TO SELECT DIFFICULT SUPPORT SETS

The first step to understanding the limitations of current few-shot classification methods is to study the difficult tasks on which they fail or perform poorly. We define a difficult task as one for which a given few-shot classifier, after being adapted to its support set, performs significantly worse on its query set compared to the mean query performance over all tasks. Finding a support set from a given search pool that leads to poor performance on a given query set for the purposes of study, however, has combinatorial complexity. Current solutions have therefore turned to greedy search-based algorithms (Agarwal et al., 2021; Fu et al., 2022), however, the computational cost quickly becomes infeasible for larger search pools and support set sizes, thus limiting study to small-scale datasets (e.g. CIFAR-FS (Bertinetto et al., 2019), mini-ImageNet (Vinyals et al., 2016)). We specifically address this limitation by proposing a fast and general optimization-based algorithm – FASTDIFF-SEL which offers a speedup of at least 20-25x over greedy search-based algorithms, allowing the extraction of difficult few-shot tasks to be scaled to a wide array of large-scale vision datasets.

### 3.1 PROPOSED METHOD

**Overview**. We present FASTDIFFSEL, that can sample a difficult few-shot classification task in a deterministic way. The key intuition of our approach is use a model's loss on the task's query set (i.e. after the model has been adapted to the support set) as a proxy for the difficulty of the task. This follows Arnold et al. (2021); Dhillon et al. (2019a) which show that a task's query loss is an effective surrogate for task difficulty and is also simple to compute. Given a model, a fixed query set and a pool of examples, we cast support set extraction as a constrained optimization problem and learn a selection weight for each example in the pool such that the loss on the query set is maximized. We can then construct a difficult support set by drawing the examples with the highest selection weights. The extracted support set is then paired with the fixed query set to form the difficult task. We can repeat this optimization for any number of query sets to extract any number of difficult tasks.

Formally, given a dataset $\mathcal{D}$, we first sample $N$ unique classes and a query set $\mathcal{Q}$. Here $Q = \{(x_r, y_r)\}_{r=1}^{N \times q}$, where $x$ is the input image and $y$ is the class label. Let $\mathcal{D}' \subset \mathcal{D}$ denote a sub-dataset containing examples from only the $N$ sampled classes and let $\mathcal{P} = \mathcal{D}' - \mathcal{Q}$ denote the set of examples from $\mathcal{D}'$ without the query set $\mathcal{Q}$. Allowing $\mathcal{P}$ to be the search pool from which the difficult support set will be extracted, the goal of the extraction algorithm is to find a support set $\mathcal{S} \subset \mathcal{P}$ such that the loss on the query set $\mathcal{Q}$ is maximized after the base model $f_\theta$ has been adapted on $\mathcal{S}$. To this end, we assume selection weights $\mathbf{w} \in \mathbb{R}^M$, where $w^i$ is associated with the $i^{\text{th}}$ example in $\mathcal{P}$ and $M = |\mathcal{P}|$ denotes the cardinality of $\mathcal{P}$. The optimization objective is to learn the selection weights $\mathbf{w}$ which result in the maximal loss for query set $\mathcal{Q}$ with a sparsity constraint on the weights. Formally:

$$\max_{\mathbf{w}} \sum_{r=1}^{N \times q} \ell((x_r, y_r), P, \mathbf{w}, f_\theta) \tag{1}$$

$$s.t. \quad w^i \in \{0, 1\}, \qquad \forall i \in [1, M]$$
$$\|\mathbf{w}_j\|_0 \leq k_j, \quad \forall j \in [1, N]$$

where $f_\theta$ is the base model trained with either meta-learning or supervised learning, $\ell$ is the loss after adapting $f_\theta$ on $\mathcal{P}$ where each of its examples are weighted by $\mathbf{w}$, and $\mathbf{w}_j$ is the selection weight vector corresponding to the $j^{\text{th}}$ class. Here $\mathbf{w} = \mathbf{w}_1 \oplus \mathbf{w}_2 \oplus ... \oplus \mathbf{w}_N$ and $w_j^i$ is the selection weight for the $i^{\text{th}}$ example in the weight vector $\mathbf{w}_j$ for the $j^{\text{th}}$ class. Note $\mathbf{w}$ are the only learnable parameters. The optimization constraints ensure that each selection weight is either 0 or 1, and that a maximum of $k_j$ examples are selected for each $j^{\text{th}}$ class. Different approaches can be used to adapt $f_\theta$ (Requeima et al., 2019; Chen et al., 2020). In our work, we adopt a ProtoNets adaptation (Snell et al., 2017) as it is highly efficient and has no learnable parameters. This approach computes a mean embedding for each class, with the loss based on the Euclidean distance between a query image embedding to each of the class prototypes.

We solve Equation (1) in two steps: (i) first, we take 1 gradient ascent step on the selection weights $\mathbf{w}$ to obtain $\hat{\mathbf{w}}$; (ii) second, we project the selection weight vector of each class $\hat{\mathbf{w}}_j$ to the $\ell_0$ norm ball to obtain the final selection weights $\bar{\mathbf{w}}_j$ ($\forall j \in [1, N]$). In practice, (ii) is known to be difficult to solve as it is NP-hard and the $l_0$ norm constraint is non-convex (Candes et al., 2005; Candes & Tao, 2005; Natarajan, 1995; Donoho, 2006). We, therefore, relax the $\ell_0$ norm to an $\ell_1$ norm to make the constraint convex following Donoho (2006) which shows that the $\ell_1$ norm relaxation gives effective sparse solutions. The projection step with $\ell_1$ relaxation for the $j^{\text{th}}$ class can be formalized as follows:

$$\min_{\mathbf{w}_j} \frac{1}{2} \|\mathbf{w}_j - \hat{\mathbf{w}}_j\|_2^2$$
$$s.t. \quad \|\mathbf{w}_j\|_1 \leq k_j \tag{2}$$

We solve the dual form of the above projection step via Lagrange multipliers (Boyd & Vandenberghe, 2004) to obtain the optimal sparse weight vector $\bar{\mathbf{w}}_j$:

$$\bar{\mathbf{w}}_j = \arg\max_{\lambda_j \geq 0} \min_{\mathbf{w}_j} \underbrace{\frac{1}{2} \|\mathbf{w}_j - \hat{\mathbf{w}}_j\|_2^2 + \lambda_j(\|\mathbf{w}_j\|_1 - k_j)}_{g(\lambda_j, \mathbf{w}_j)} \tag{3}$$

where $\lambda_j$ is the Lagrange multiplier corresponding to the projection step for the $j^{\text{th}}$ class. In practice, we solve the projection step per class to ensure at least a few-examples per class are selected, and we find that 1 projection step per class is sufficient to learn appropriate selection weights. After 1 gradient ascent and 1 projection step (i.e. weight vectors have been learned), we select the examples from $\mathcal{P}$ to include in the difficult support set. For each $j^{\text{th}}$ class, we sort the final selection weight vector $\bar{\mathbf{w}}_j$ in descending order and extract the $k_j$ examples from $\mathcal{P}$ which have the highest weights.

The pseudo-code for FASTDIFFSEL is shown in Algorithm 1 and a detailed derivation of the steps for solving the optimization along with the associated hyperparameters can be found in Appendix A.2.

**Computational complexity**. The key advantage of FASTDIFFSEL is that it does not require an iterative exhaustive search through the search pool to select a difficult support set. Consider selecting a task with $N$ classes and $k$ support examples per class from a dataset $\mathcal{D}'$, where each class has $d$ examples on average. The greedy algorithm in Agarwal et al. (2021) runs for $r$ iterations and thus

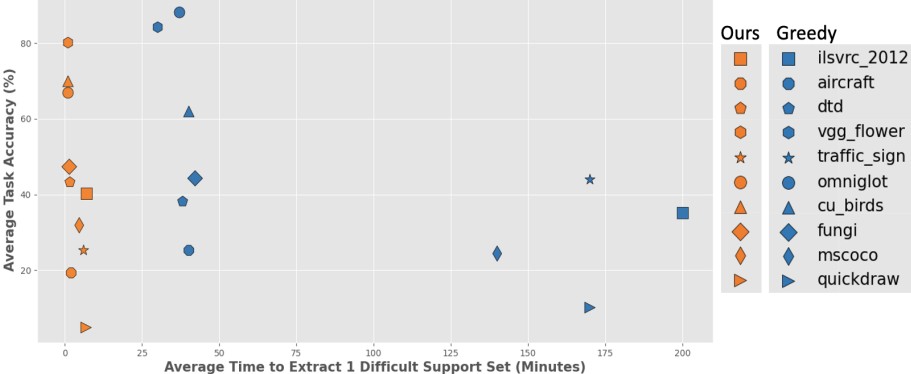

Figure 2: **FASTDIFFSEL extracts tasks with similar accuracy to those extracted by greedy search algorithms (Agarwal et al., 2021) but is at least 20x faster**. We extract 50 5-way 5-shot tasks per sub-dataset using Hu et al. (2022) as the base model (ViT-S initialized with SSL DINO weights then meta-trained with ProtoNets on MD's ilsvrc_2012) on an A5000 GPU (64GB RAM).

has a search time complexity of $\mathcal{O}(N.k.d.r)$ along with as many adaptation steps. In comparison, our algorithm removes the need for this exhaustive search and large number of adaptation steps, thus offering a significant speedup. In practice, we find that our algorithm offers speedups of at least 20x when compared to a greedy algorithm (see Fig 2 and Fig 6).

## 4 DIFFICULT SUPPORT SET EXTRACTION ON META-DATASET

Since FASTDIFFSEL is general and highly efficient, we can use it to extract difficult tasks from any large-scale vision dataset. MD is one of the most widely-used benchmarks for few-shot classification, thus we primarily use it to validate the effectiveness of our extraction algorithm. We include further analyses on difficult tasks extracted from further datasets in Section 4.2.

**META-DATASET** (MD) (Triantafillou et al., 2019) contains 10 image sub-datasets from a diverse set of domains. All the sub-datasets (except mscoco and traffic_signs) are split into disjoint train/val/test classes, whereas mscoco and traffic_sign are test only sub-datasets. During few-shot training, variable-way variable-shot tasks are randomly sampled from the train classes of each sub-dataset. During few-shot testing, 600 variable-way variable-shot tasks are sampled from the test classes of each sub-dataset with the average task classification accuracy reported for each sub-dataset.

### 4.1 TEST TASK SAMPLERS FOR META-DATASET

We compare tasks sampled from the test split of MD's sub-datasets using 3 methods: (i) MD's default sampler; (ii) the greedy approach of Agarwal et al. (2021) and (iii)FASTDIFFSEL(Ours). Note, only (ii) and (iii) can be used to deterministically sample difficult tasks:

**MD's default sampler.** The default task sampler in MD samples imbalanced tasks of variable-way, variable-shots. The shot, way and size of the query set depends on the size of the sub-dataset in MD. For more details, refer to Sec 3.2 in Triantafillou et al. (2019). We note, however, that the default sampler cannot deterministically sample difficult tasks.

**Greedy search-based sampler.** We use Agarwal et al. (2021) to extract difficult tasks from MD. This approach works by iteratively replacing each example in a support set of fixed size with one from a given search pool such that the query loss is maximized. Because of the large computational cost and time associated with searching the pool each time an example is replaced, greedy approaches do not scale well to MD's sampler where task ways can vary as large as 50 and shots as large as 100. We therefore only consider fixed-way, fixed-shot tasks when sampling via greedy search on MD.

**FASTDIFFSEL.** We use FASTDIFFSEL to extract difficult tasks from MD. As the trained base model $f_\theta$, we choose a state-of-the-art method (Hu et al., 2022) on MD which employs a ViT-S feature extractor (Touvron et al., 2020) initialized with self-supervised DINO weights (Caron et al., 2021) pre-trained on ilsvrc_2012. The extractor is then further meta-trained on the ilsvrc_2012 split from MD. To compare against the above samplers, we consider two configurations:

- **Variable-way, variable-shot tasks**. Here, we exactly match the variable way and shots per class of each task yielded by MD's default sampler. Specifically, we use the default sampler to generate and save the shot and way for a fixed number of tasks, which we then feed into our algorithm to extract difficult tasks of equivalent specifications.

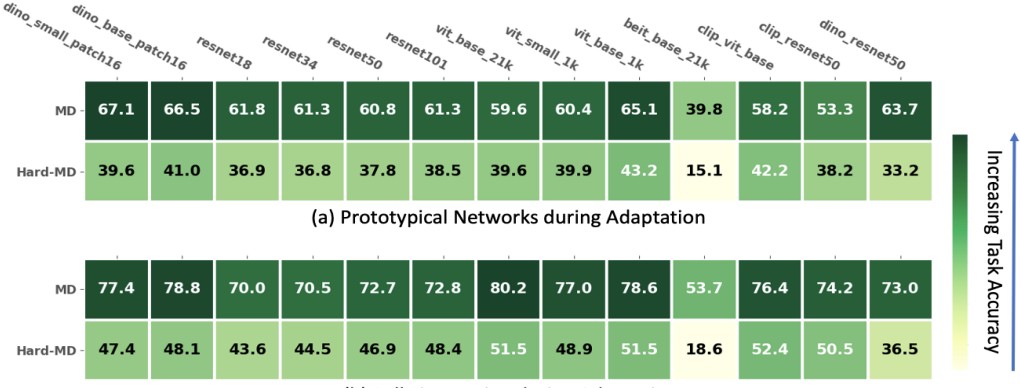

Figure 3: **We evaluate a wide range of top-performing methods over 200 variable-way, variable-shot difficult tasks per sub-dataset in HARD-MD and find a consistent 20-30% drop in performance compared to MD tasks regardless of feature extractor, pretraining method, and adaptation strategy.** We evaluate on all MD sub-datasets except ilsvrc_2012. We include performance on each sub-dataset and their 95% confidence intervals in Appendix H.

- **Fixed-way, fixed-shot tasks**. Here, we set the way and shots per class to be fixed for all tasks. While less challenging, it allows us (i) to compare our algorithm to greedily sampled difficult tasks; and (ii) to control for a task's way being a source of difficulty.

## 4.2 VALIDATION OF DIFFICULT META-DATASET TASKS

Here we compare the efficacy of the above 3 methods and their extracted tasks across the 10 test sub-datasets in MD.

**FASTDIFFSEL vs. default sampler.** When compared over 600 variable-way variable-shot tasks per sub-dataset, we find that our algorithm extract tasks which are more difficult on an average than MD's default sampler, as shown in Fig 1. This is consistent across all 10 sub-datasets. We also show that these tasks are consistently difficult for a wide range of top-performing few-shot classification methods, as shown in Fig 3 (see Section 5.3.1 for details). For certain sub-datasets (e.g. quickdraw), the drop in classification accuracy can be as large as 50% when compared to tasks sampled from MD's default sampler.

**FASTDIFFSEL vs. greedy sampler.** We compare our algorithm to the greedy sampler over 50 5-way 5-shot tasks sampled per sub-dataset. For fairness, we run both methods for the same set of query sets, and report the average task accuracy and extraction time in Fig 2. Here, we see that the average task accuracy is comparably low for each sub-dataset but that our algorithm is almost 1-2 orders of magnitude faster. This speedup is most significant when the search pool is large (e.g. for ilsvrc_2012, quickdraw). Note that the greedy sampler takes between 20-200 minutes *per task*, highlighting its impracticality for larger search pools, support set sizes and numbers of tasks.

In all, FASTDIFFSEL is able to consistently extract difficult tasks on which state-of-the-art methods achieve low classification accuracy, while also offering a significant speedup in extraction time compared to existing task samplers. It can, therefore, readily be leveraged to further the study of failure modes in few-shot classification methods.

## 5 STRESS TESTING WITH HARD-META-DATASET++

Few-shot classification benchmarks like MD (Triantafillou et al., 2019) and VTAB+MD (Dumoulin et al., 2021b) are highly challenging but are not specifically geared to driving performance on difficult tasks. We leverage our extraction algorithm to fill this gap and introduce HARD-MD++, a new test-only benchmark of exclusively difficult few-shot classification tasks extracted from 4 large-scale vision datasets. We extensively stress test a wide range of top-performing methods on HARD-MD++, presenting novel insights on the robustness of current methods to difficult tasks.

## 5.1 TEST DATASETS

HARD-MD++ is composed of difficult tasks extracted from MD (Triantafillou et al., 2019), OR-BIT (Massiceti et al., 2021), CURE-OR (Temel et al., 2018) and OBJECTNET (Barbu et al., 2019). We motivate the choice of MD by the fact that it is one of the most widely-used few-shot classification benchmarks. The remaining datasets are chosen because they specifically curate images with real-world variations and provide corresponding 'quality' annotations. They can, therefore, be lever-

aged to derive deeper insights into the *properties* of difficult tasks as we explore in Section 5.3.2. Together, these datasets cover a broad range of challenging and real-world tasks.

Following Section 4, we use FASTDIFFSEL to extract 200 difficult tasks from each dataset. We use a trained base model $f_\theta$ of ViT-S (Dosovitskiy et al., 2020) pre-trained using DINO (Caron et al., 2021) and further meta-trained on MD's ilsvrc_2012 split. For MD, we extract variable-way variable-shot tasks to align with its existing evaluation protocol. For the remaining datasets, we extract fixed-way fixed-shot tasks to enable controlled analysis on task properties beyond their way and shot. Together, HARD-MD++ comes to a total of 2400 difficult test tasks extracted across these datasets. We include further implementations details in Appendix B.

**META-DATASET.** We extract 200 variable-way variable-shot difficult tasks from each test sub-datasets in MD. Note, we exclude the ilsvrc_2012 subset so as not to prevent the use of feature extractors pre-trained on it.

**ORBIT** contains 3822 videos of 486 objects captured by people who are blind on their mobile phones. Each frame is annotated with 7 quality issues including blur, framing, lighting and occlusion. We extract 200 fixed 5-way, 5-shot difficult tasks from the test split of ORBIT which cover 1198 videos of 158 objects.

**CURE-OR** contains 1M test images of 100 objects captured under controlled viewpoint variations (e.g., front, back, side etc.) with corresponding annotations. We extract 200 fixed 5-way, 5-shot difficult tasks from CURE-OR.

**OBJECTNET** contains 50K test images of 313 objects captured under rotation, background, and viewpoint variations. We extract 200 fixed-way fixed shot difficult tasks from OBJECTNET.

## 5.2 METRICS AND TRAINING

**Metrics**. Model performance on HARD-MD++ should be reported as the average classification accuracy and 95% confidence interval over the difficult tasks per sub-dataset. This should be accompanied by performance on MD to provide a more complete characterization of model performance.

**Training**. We primarily advocate for cross-domain or strong generalization (Triantafillou, 2021), thus following Hu et al. (2022), any pre-trained checkpoint, algorithm and model architecture can be used when evaluating on HARD-MD++. This aligns with more recent few-shot learning practices which allow the wide range of publicly-available datasets and pre-trained models to be leveraged.

## 5.3 RESULTS

HARD-MD++ is a challenging benchmark of difficult tasks from a diverse range of datasets. We stress test a wide range of top-performing few-shot classification methods on HARD-MD++ to gauge their robustness to these tasks. We look first at MD and then ORBIT, CURE-OR, and OBJECTNET, primarily focusing on the robustness of different model architectures and pre-training strategies.

**State-of-the-art methods.** Recent works (Tian et al., 2020; Hu et al., 2022; Dhillon et al., 2019b; Chen et al., 2019; Dumoulin et al., 2021a) show that transfer learning with a powerful feature extractor performs extremely well on few-shot classification tasks compared to previous meta-learning methods (Finn et al., 2017a; Requeima et al., 2019). We therefore consider a wide range of feature extractors and pre-training paradigms: ResNets (He et al., 2015), Vision Transformers (ViTs) (Dosovitskiy et al., 2020; Touvron et al., 2020), self-supervised variants of ResNets and ViTs (Caron et al., 2021; Bao et al., 2021) and the visual encoder of zero-shot models such as CLIP (Radford et al., 2021). We investigate two adaptation strategies when adapting to each difficult test task in HARD-MD++: (i) computing and classifying by mean class prototypes following Prototypical Networks (Snell et al., 2017) and (ii) fine-tuning the entire feature extractor using strong augmentations following (Hu et al., 2022). Further details are provided in Appendix E.

### 5.3.1 RESULTS ON DIFFICULT TASKS FROM META-DATASET

Our key finding is that state-of-the-art methods across the board consistently drop at least 20-25% in classification accuracy when adapting to difficult tasks in HARD-MD compared with tasks randomly sampled from MD, shown in Fig 3. Note, HARD-MD refers to the MD split in HARD-MD++.

**Detailed findings.** We find that there is a consistent drop in classification accuracy on HARD-MD across all state-of-the-art methods, regardless of adaptation strategy, feature extractor and pre-training paradigm (see Fig 3). This validates our proposed algorithm in its ability to extract generally challenging tasks, but also highlights the limitations of current approaches. In particular, approaches

| | dino_small_patch16 | dino_base_patch16 | resnet18 | resnet34 | resnet50 | resnet101 | vit_base_21k | vit_small_1k | vit_base_1k | beit_base_21k | clip_vit_base | clip_resnet50 | dino_resnet50 |
|---|---|---|---|---|---|---|---|---|---|---|---|---|---|
| ObjectNet | 13.8 | 17.8 | 17.3 | 18.7 | 20.1 | 19.3 | 23.0 | 27.2 | 28.0 | 16.5 | 43.6 | 27.9 | 13.2 |
| CURE-OR | 24.7 | 25.3 | 27.0 | 27.0 | 28.1 | 26.8 | 27.8 | 27.9 | 27.2 | 13.2 | 32.8 | 26.9 | 18.7 |
| ORBIT | 22.5 | 25.6 | 30.2 | 30.3 | 32.2 | 34.7 | 35.1 | 39.8 | 40.4 | 16.7 | 38.2 | 32.3 | 25.0 |

Figure 4: **We stress test a wide range of few-shot classifiers on difficult tasks from OBJECTNET, CURE-OR and ORBIT in HARD-MD++.** We find that ViT-B (CLIP) outperforms all the other models by a large margin for CURE-OR and OBJECTNET, while being extremely competitive for ORBIT. Evaluation across 200 tasks per domain using Prototypical Networks as the adaptation strategy. More results with fine-tuning in Appendix H.

that employ a fine-tuning adaptation strategy see a more significant drop in accuracy compared to those employing a prototypical-style adaptation. For example, fine-tuning a ViT-B feature extractor pre-trained on ImageNet-21k ('vit_base_21k' in Fig 3) (Dosovitskiy et al., 2020) on each test task leads to a drop of ∼30% in accuracy on HARD-MD while a prototypical-style adaptation leads to a drop of only ∼20% (note, this approach is strongest on original MD in our implementation). This suggests that although fully fine-tuning a model can significantly increase its accuracy on general tasks, it may not hold for specifically difficult tasks. On the other hand, we find the visual encoder of CLIP which also uses a ViT-B architecture ('clip_vit_base' in Fig 3) (Radford et al., 2021) to be more robust to difficult tasks. Despite under-performing on MD (ranks $6^{\text{th}}$), it displays strong performance on HARD-MD across both adaptation strategies, achieving the highest accuracy across all methods with fine-tuning. This trend is consistent for fixed-way, fixed-shot tasks on HARD-MD (see Appendix H). These early results suggest that large-scale vision-language pre-training may offer more robustness when generalising to new difficult tasks.

**Effect of meta-training on ilsvrc_2012.** Hu et al. (2022) show that a method's performance on MD can be improved by further meta-training its pre-trained feature extractor on MD's ilsvrc_2012 split. We investigate whether this can also improve performance on HARD-MD. We further meta-train a subset of the methods on the ilsvrc_2012 split using Prototypical Networks (Snell et al., 2017) and compare the average task classification accuracy in Appendix H. We find that despite the further meta-training, the tested methods still display a consistent drop of 20-30% on HARD-MD. This suggests that novel algorithmic contributions, rather than further training, may be required to improve robustness to difficult tasks.

### 5.3.2 RESULTS ON DIFFICULT TASKS FROM CURE-OR, ORBIT AND OBJECTNET

Similar to Section 5.3.1, our key finding is that state-of-the-art methods consistently achieve low classification accuracy on difficult tasks from CURE-OR, ORBIT and OBJECTNET, shown in Fig 4. In particular, ViT-B pre-trained with CLIP (Radford et al., 2021) outperforms all other methods by a significant margin on both CURE-OR (by >4%) and OBJECTNET (by >15%). On ORBIT, ViT-B pre-trained on ilsvrc_2012 (Deng et al., 2009) performs best though ViT-B with CLIP is still competitive.

We can go beyond task classification accuracy with CURE-OR and ORBIT by leveraging their per-image annotations to investigate the *properties* of difficult tasks. In particular, we use the object viewpoint annotations in CURE-OR, and all quality issue annotations (except object_not_present) in ORBIT. We compare difficult tasks extracted by our algorithm for each dataset with 'easy' tasks for a fixed 5-way 5-shot setting. We extract 'easy' tasks by instead *minimizing* the objective in Equation (1) for the same set of query sets. In Fig 5, we compare the composition of these difficult and easy tasks by the annotated attributes in their supports sets.

**CURE-OR.** We randomly sample 200 query sets containing images only showing the object with a front or back viewpoint. The sampled query sets contain 10 examples per class. For each query set, we use our algorithm to extract an easy and a difficult support set and visualize the distribution of their annotations in Fig 5-(Left). Here, we see that the easy support sets have a higher proportion of images with front or back viewpoints relative to the total size of the support set, while the difficult support sets have a significantly higher proportion of images with side viewpoints.

**ORBIT.** Unlike CURE-OR, ORBIT was not collected in a controlled setting and hence its quality issue annotations have a long-tailed distribution. We therefore randomly sample 200 query sets, and for each, we use FASTDIFFSEL to extract an easy and a difficult support set. For each quality issue, we compute the difference (gap) in the proportion of images in the support versus query set with that

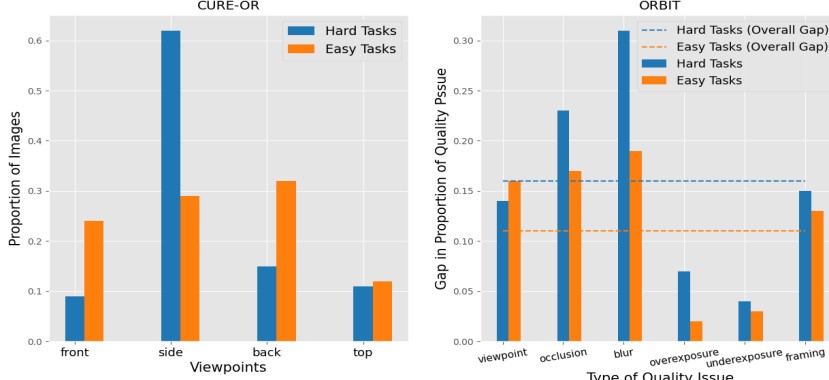

Figure 5: **Difficult few-shot tasks have a larger mismatch between the properties of their support and query sets compared to easy tasks.** (Left): Difficult tasks in CURE-OR have support sets with the majority of images showing the object in side view when the query sets contain images of the object only in front/back view; (Right): Difficult tasks in ORBIT have a larger difference in the proportion of quality issues contained in their support versus query sets compared to easy tasks.

particular issue. In Fig 5-(Right), we report this gap averaged over 200 tasks. Here, we observe that difficult tasks have a larger gap (i.e. difference in support and query set) in quality issues than easier tasks. Together, these results suggest that a mismatch between image characteristics within a support versus query set (e.g., viewpoint, occlusion) can be a source of task difficulty. While these annotations do not cover all possible image characteristics, they provide a starting point to explore the robustness of few-shot classifiers to more qualitative distribution shifts. In Appendix C, we curate few-shot tasks leveraging image-level annotations from ORBIT, CURE-OR and MS-COCO – to show that tasks with distribution shifts with respect to natural characteristics have a lower accuracy.

## 6 RELATED WORKS

**Difficult tasks.** Previous works (Agarwal et al., 2021; Arnold et al., 2021; Dhillon et al., 2019a) have shown that state-of-the-art few-shot classifiers generally display a wide range in performance when adapting to different test tasks. Agarwal et al. (2021) use this observation to develop a greedy search-based algorithm that can specifically extract difficult tasks for further study. They consider only meta-learning approaches and, due to the computational requirements of a greedy search, are limited to small-scale datasets including mini-ImageNet and CIFAR-FS. (Arnold et al., 2021) also study difficult tasks through a correlation-based analysis. We extend on both of these works by (i) proposing a scalable algorithm – FASTDIFFSEL that can extract difficult tasks from any large-scale vision dataset, and (ii) conducting a deep empirical evaluation on the robustness of a broader range of meta-learning and transfer learning approaches on these difficult tasks. Potentially ideas from subset selection (Wei et al., 2015; Killamsetty et al., 2020) can be adapted for few-shot task extraction, but we leave it for future work.

**Few-shot classification benchmarks.** MD (Triantafillou et al., 2019) and VTAB+MD (Dumoulin et al., 2021b) are two of the most challenging few-shot image classification benchmarks in the current literature. They cover a wide range of domains and primarily evaluate the ability of a few-shot classifier to generalise to novel object classes, datasets and domains. Other few-shot benchmarks have been introduced to specifically target adaptation to images with high real-world variation, including ORBIT (Massiceti et al., 2021) and cross-domain transfer beyond natural images, including BSCD-FSL (Guo et al., 2020). We note, however, that unlike HARD-MD++, none of these benchmarks specifically target difficult tasks for few-shot classification.

## 7 CONCLUSION

We introduce a general and scalable algorithm – FASTDIFFSEL to extract difficult tasks for few-shot classification. We apply FASTDIFFSEL to 4 large-scale vision datasets: META-DATASET, ORBIT, CURE-OR, and OBJECTNET, and introduce HARD-MD++, a new *test-only* suite of 2400 difficult tasks from across these datasets. We stress test a wide range of top-performing few-shot methods on HARD-MD and demonstrate a consistent drop of 20-25% in classification accuracy compared to MD. We conduct additional quantitative analyses on CURE-OR and ORBIT which show that difficult tasks typically have a distribution shift in the characteristics between a support and query set (e.g. viewpoint, blur). We believe that the efficiency of FASTDIFFSEL along with HARD-MD++ can drive the study of failure modes in few-shot classification methods which is a under explored area of research.

## 8 ACKNOWLEDGEMENTS

This project was supported in part by Meta grant 23010098, NSF CAREER AWARD 1942230, HR001119S0026 (GARD), ONR YIP award N00014-22-1-2271, Army Grant No. W911NF2120076 and the NSF award CCF2212458.

## 9 REPRODUCIBILITY STATEMENT

Our work primarily consists of (i) an algorithm – FASTDIFFSEL for extracting difficult few-shot tasks from any vision dataset. (ii) a few-shot testing benchmark called HARD-MD++. For (i), we provide the steps for reproducing the algorithm in Algorithm 1. The hyper-parameters for the algorithm can be found in Appendix A.2. For (ii), we provide all the relevant details in Appendix B and Section 5. The details for training and fine-tuning the various models that we stress test on HARD-MD++ can be referred in Appendix F. We will publically release the code and HARD-MD++ upon the acceptance of our manuscript.

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

# A  FASTDIFFSEL: SUPPORT SET EXTRACTION ALGORITHM

## A.1  STEPS FOR SOLVING THE PROJECTION STEP

In this section, we provide details on how to solve the projection step in Equation (3). The projection step is solved separately for each of the $j^{th}$ class, where $j \in [1, N]$. $\hat{\mathbf{w}}_j$ is the selection weight vector for the $j^{th}$ class obtained after a step of gradient ascent on Equation (1). The dual form of Equation (2) can be expressed via Lagrange multipliers as the following:

$$\bar{\mathbf{w}}_j = \arg\max_{\lambda_j \geq 0} \min_{\mathbf{w}_j} \underbrace{\frac{1}{2}\|\mathbf{w}_j - \hat{\mathbf{w}}_j\|_2^2 + \lambda_j(\|\mathbf{w}_j\|_1 - k_j)}_{g(\lambda_j, \mathbf{w}_j)} \tag{4}$$

We solve Equation (4) in two steps: (i) First, we solve $\min_{\mathbf{w}_j} g(\lambda_j, \mathbf{w}_j)$ via proximal operators; (ii) Then, we obtain the optimal values of the dual parameters $\lambda_j$.

KKT optimality conditions (due to stationarity) states that $\nabla_{\mathbf{w}_j} g(\lambda_j, \mathbf{w}_j) = 0$. However, note that $g(\lambda_j, \mathbf{w}_j)$ is a combination of a smooth function and a non-smooth function which can be solved by proximal operators. Considering $\mathbf{w}_j \in \mathbb{R}^n$, the KKT optimality condition can be stated as the following:

$$\nabla_{\mathbf{w}_j} \frac{1}{2}\|\mathbf{w}_j - \hat{\mathbf{w}}_j\|^2 + \nabla_{\mathbf{w}_j} \lambda_j(\|\mathbf{w}_j\|_1 - k_j) = 0 \tag{5}$$

The value in the $i^{th}$ index in $\mathbf{w}_j$, which is $w_j^i$ can be obtained through:

$$\frac{1}{2}\frac{\partial(w_j^i - \hat{w}_j^i)^2}{\partial w_j^i} + \lambda_j \frac{\partial|w_j^i|}{\partial w_j^i} = 0 \tag{6}$$

If $w_j^i > 0$, then the derivative in Equation (6) is $w_j^i - \hat{w}_j^i + \lambda_j$. Therefore Equation (6) can be expressed as: $\bar{w}_j^i = \hat{w}_j^i - \lambda_j$, which holds true for $\hat{w}_j^i > \lambda_j$. Similarly, when $w_j^i < 0$, $\bar{w}_j^i = \hat{w}_j^i + \lambda_j$ is the minimizer. For $\hat{w}_j^i \in [-\lambda_j, \lambda_j]$, the minimizer is at the only point of differentiability for which $w_j^i = 0$. This operation is called soft-thresholding and can be expressed as:

$$\bar{w}_j^i = \text{Prox}_{\lambda^*\|.\|_1}(\hat{w}_j^i) = \text{sign}(\hat{w}_j^i)\max(|\hat{w}_j^i| - \lambda_j^*, 0) \tag{7}$$

Thus, $\bar{\mathbf{w}}_j = \text{Prox}_{\lambda^*\|.\|_1}(\hat{\mathbf{w}}_j) = [\text{Prox}_{\lambda^*\|.\|_1}(\hat{w}_j^1), ...., \text{Prox}_{\lambda^*\|.\|_1}(\hat{w}_j^n)]$. The next step is to compute the value of the dual parameter $\lambda_j^*$. We then compute the derivative $g'(\lambda_j, \bar{\mathbf{w}}_j)$ as the following:

$$g'(\lambda_j, \bar{\mathbf{w}}_j) = \|\text{Prox}_{\lambda^*\|.\|_1}(\hat{\mathbf{w}}_j)\|_1 - k_j \tag{8}$$

$$= \sum_{i=1}^{n}(|\hat{w}_j^i| - \lambda_j)_+ - k_j \tag{9}$$

We solve Equation (9) by the root finding method in (Corliss, 1977), since the optimal $\lambda_j^* \in [0, \|\hat{\mathbf{w}}_j\|_\infty]$. The upper bound $\|\hat{\mathbf{w}}_j\|_\infty$ ensures that $g'(\lambda_j, \bar{\mathbf{w}}_j)$ does not become negative.

## A.2  HYPERPARAMETERS OF FASTDIFFSEL

Empirically, we perform a grid-search for the learning rate $\alpha \in [0.01, 500]$ with a step size of 10. We specifically find that a high learning rate with *only* one gradient ascent and projection step suffices to obtain difficult tasks. In our experiments, we use a learning rate $\alpha = 200$, as we find this value to result in the extraction of the most difficult tasks.

## A.3  EMPIRICAL RUNNING TIMES

In Fig. (6), we plot the average running time of our framework to extract a single difficult support set across all the domains in META-DATASET. In practice, we find a large speedup (at least 20x) using our framework over (Agarwal et al., 2021) for support set extraction. The main advantage of our framework is the elimination of the iterative search in (Agarwal et al., 2021), which is computionally expensive. The speedup is crucial for scalability of difficult task extraction to large-scale vision datasets such as MD.

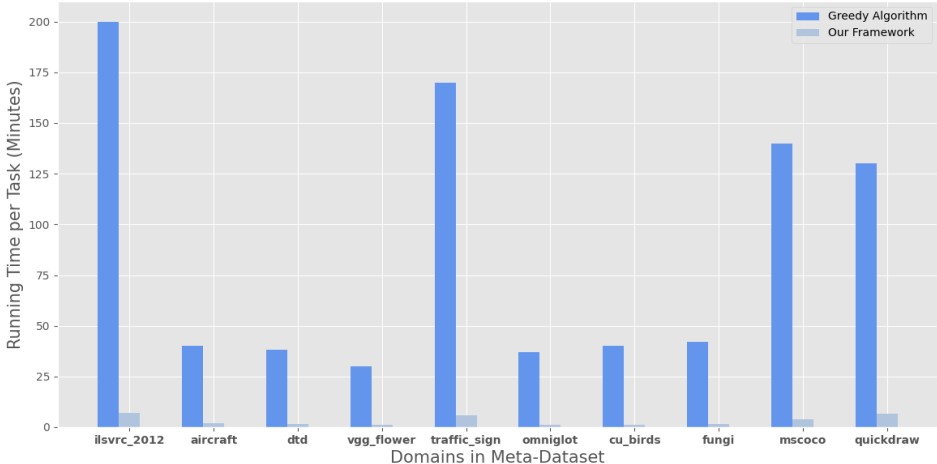

Figure 6: **Average running times of our framework (In practice)**: For extracting a difficult task on an average, we find a speedup of at least 20x (depending on the dataset) when compared to the greedy method in (Agarwal et al., 2021). Tasks are fixed-way, fixed-shot tasks where way=5, shot = 5. On hardware with a5000 GPU with 64GB RAM.

## B  DESIGN CHOICES FOR DATASETS

**Choice for META-DATASET.** META-DATASET(Triantafillou et al., 2019) is arguably one of the most challenging few-shot learning benchmarks comprising of 10 different sub-datasets spanning different domains. We primarily build HARD-MD++ on top of MD, to enable fast adoption of our benchmark by the few-shot learning community. We envision HARD-MD++ to be an add-on to MD, which the community uses to report performance on difficult tasks.

**Choice for CURE-OR and ORBIT.** MD though a challenging few-shot learning benchmark, does not come with image-level annotations which can be leveraged to understand the properties of difficult tasks. Understanding the properties of difficult tasks is crucial, as they can further guide the development of future algorithmic methods specifically for tackling difficult few-shot tasks. We ground our choice of annotated datasets in CURE-OR(Temel et al., 2018) and ORBIT(Massiceti et al., 2021), as they reflect the variations present in real-world images. While ORBIT is an existing few-shot learning benchmmark comprising of real-world videos of everyday objects, CURE-OR is primarily an object recognition dataset. In CURE-OR, each image has objects in 5 different viewpoints, which are curated in a controlled environment. ORBIT consists of 7 unique annotations about various quality issues for each frame in the video. These annotations are blur, viewpoint, framing, occlusion, overexposure, underexposure and object_not_present. In our experiments, we specifically leverage all the quality annotations except object_not_present. These issues are reflective of a subset of properties which can arise in real-world images. We use these annotations from ORBIT and CURE-OR to understand the properties of difficult few-shot tasks. In Section 5.3.2, we provide some early analyses in this regard, where we find that difficult tasks have a distribution shift between the support and query set, with respect to various image-level properties such as blur, viewpoint etc. Methodological evaluation on CURE-OR and ORBIT, will enable researchers to understand the specific dimensions of image characteristics in which in which robustness has improved.

**Choice for OBJECTNET.** A commonly used high-variation test-set in supervised deep learning is OBJECTNET, which is often used for testing the out-of-domain generalization capabilities of models. In particular OBJECTNET comprises of images of objects in various viewpoints, backgrounds and rotations which results in high variation within a class of objects. However, we note that these annotations are not made public by the (Barbu et al., 2019). Considering the ability of our algorithm to extract tasks with a distribution shift between support and query with respect to various image characteristics in high variation datasets such as ORBIT and CURE-OR, we extract difficult tasks from OBJECTNET and add them to our benchmark HARD-MD++.

As the next step, the difficult tasks from the MD split in HARD-MD++ can be annotated similar to CURE-OR or ORBIT to furthen our understanding of the properties of difficult few-shot tasks.

## C  WHAT CAN MAKE A FEW-SHOT TASK DIFFICULT?

Prior works (Agarwal et al., 2021; Massiceti et al., 2021; Dhillon et al., 2019a; Arnold et al., 2021) have shown that the composition of the support or query set is crucial in determining the difficulty level of a task. This difficulty level is usually characterized by the loss or accuracy on the query set. While an intuitive characterization, it gives limited insights into *why* the given few-shot task was difficult. In this section, we aim to shed light on this question by specifically focusing on the role that natural image characteristics (e.g., quality issues, multiple objects) play in making a task difficult. These natural characteristics can arise in the wild, hence we specifically focus on them. We note that there can be other factors controlling task difficulty such as fine-grained classes, however it is outside the scope of this paper (see Table 1 for more details). We hypothesize that tasks which are difficult for a given few-shot learner have a mismatch (distribution shift) between the support and the query set with respect to certain image-level characteristics. To validate our hypothesis, we curate annotated few-shot tasks where there is a mismatch in the characteristics between the support and query set. We leverage three vision datasets which are annotated with various image-level attributes to curate the few-shot tasks: (i) ORBIT (Massiceti et al., 2021) which contains annotations designating if the frame is cluttered or not as well annotations of various quality issues such as blur, occlusion, overexposure, underexposure, viewpoint and framing. (ii) CURE-OR (Temel et al., 2018) consists of images of various objects in different viewpoints. (iii) MSCOCO (Lin et al., 2014) consists of annotations about the number of objects in an image.

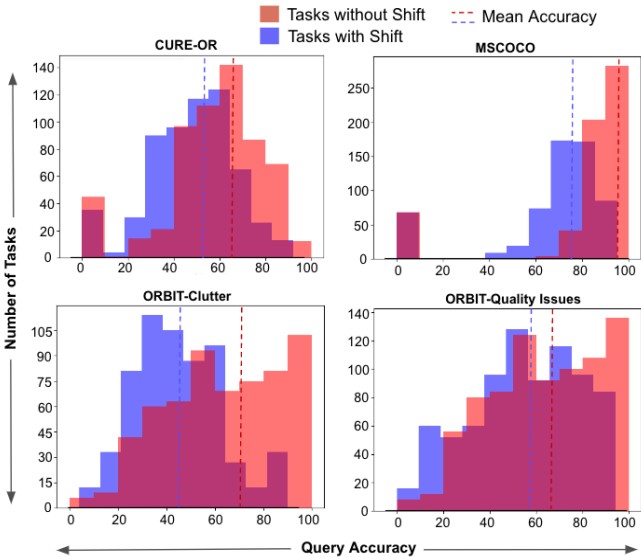

Figure 7: **Curated tasks with a distribution shift between their support and query sets with respect to image-level attributes have a lower accuracy on average**. Across 600 tasks (5-way, 5-shot) per dataset on ViT-S(DINO). Task curation details in Appendix D.

**Sampling Protocol**. For each of the annotated datasets, we curate difficult few-shot tasks in two steps: (i) We sample a query set $Q$ which contains a certain set of attributes; (ii) For curating tasks without shift, we curate a support set $S$ containing similar attributes as $Q$. For tasks with shift, the curated support set contains attributes different from those in $Q$. Further task curation details are in Appendix D.

**Results.** From Fig 7, we find that tasks that have a distribution shift between the attributes of their support versus query set have lower query accuracy on average compared to tasks that do not have this shift. We also find a strong correlation between the query accuracy and the inverse of the query

loss with a Spearman correlation of 0.90 (See Fig 8). This shows that tasks with a high query loss (low accuracy) is representative of distribution shifts between the support and query with respect to natural image characteristics.

## C.1 CORRELATION PLOTS

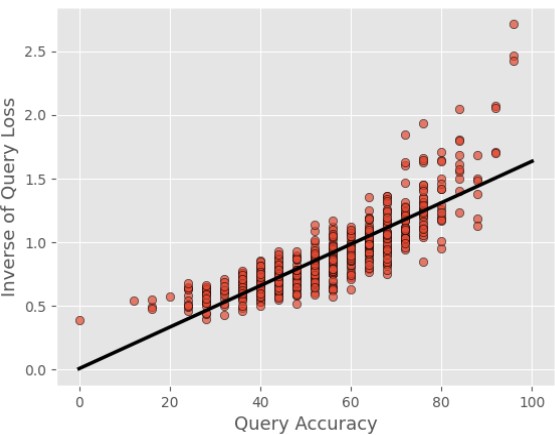

Figure 8: **Spearman correlation of 0.90 between the query accuracy and inverse of the query loss for tasks sampled from CURE-OR and ORBIT.** 600 tasks are sampled from each of the dataset.

## D HEURISTIC FEW-SHOT TASK CURATION DETAILS

In Fig 7, we heuristically curate two types of tasks leveraging attribute annotated vision datasets such as CURE-OR, ORBIT and MS-COCO: (i) Tasks with shift between the support and query set. (ii) Tasks without shift between the support and query set. Below we provide details on how these tasks are curated:

### D.1 MS-COCO

**Tasks with Shift.** We sample $N$ classes (where $N = 5$) and randomly select a query set $Q$ from the pool of images having only one object. The support set $S$ is randomly selected from the pool of examples containing more than 3 objects. Each task is a 5-way, 5-shot task, where there is a large mismatch between the number of objects present in the support vs. query.

**Tasks without Shift.** We sample $N$ classes (where $N = 5$) and randomly select a query set $Q$ from the pool of images having only one object. The support set $S$ is randomly selected from the pool of examples containing only one or two objects. Each task is a 5-way, 5-shot task, where there is no significant mismatch between the number of objects present in the support vs. query.

### D.2 ORBIT

#### D.2.1 ORBIT-CLUTTER

**Tasks with Shift.** We first randomly sample a user and $N$ classes (where $N=5$). The query set $Q$ is then selected from the clean frames. The support set corresponding the the same user and $N$ classes is then selected randomly from the cluttered frames. This sampling procedure ensures that there is a shift between the support and query with respect to the number of objects in the frame.

**Tasks without Shift.** We first randomly sample a user and $N$ classes (where $N=5$). The query set $Q$ is then selected from the clean frames. The support set corresponding the the same user and $N$ classes is then also selected randomly from the clean frames. This sampling procedure ensures that there is no shift between the support and query with respect to the number of objects in the frame.

### D.2.2 ORBIT-QUALITY ISSUES

**Tasks with Shift.** We first randomly sample a user and $N$ classes ($N = 5$) corresponding to the user. We then select a query set $Q$ which contains images without any quality issues. To create the support set, we create a subset by selecting frames which have one or more quality issues (i.e. blur==True OR occlusion==True OR overexposure==True OR underexposure==True OR viewpoint==True OR framing==True). Then, we randomly sample a support set $S$ from this subset which contains frames with quality issues. This design choice ensures that the tasks contain a significant shift between the support and query with respect to various quality issues.

**Tasks without Shift.** To create tasks without any shift, we use the default sampler from ORBIT, but modify it to select fixed-way, fixed-shot tasks with the query set $Q$ and support set $S$. Because the tasks are sampled randomly, a majority of the tasks will only have minor shifts with respect to the various quality issues.

### D.3 CURE-OR

**Tasks with Shift.** We sample $N$ classes (where $N = 5$) and randomly select a query set $Q$ which has objects in either (i) front or back viewpoints; (ii) side-viewpoints and (iii) top viewpoints. If (i) is selected, then the support set $S$ is composed of images only with side-viewpoints. If (ii) is selected, then the support set $S$ is composed of images with only the front or back viewpoints. If (iii) is selected, then the support set $S$ is composed of images with front, back or the side viewpoints. This ensures that the tasks have a shift between the support and query with respect to the viewpoint attribute.

**Tasks without Shift.** We sample $N$ classes (where $N = 5$) and randomly select a query set $Q$ which has objects in either (i) front or back viewpoints; (ii) side-viewpoints and (iii) top viewpoints.If (i) is selected, then the support set $S$ is composed of images only with the front or back viewpoints. If (ii) is selected, then the support set $S$ is composed of images only with the side viewpoints. Similarly, if (iii) is selected, the support set is composed of images containing only the top viewpoint of the object. This design decision ensures that tasks do not have any shift between the support and query with respect to the viewpoint attribute.

We note that these rules or heuristics are not exhaustive to create tasks which have a shift between the support and query set with respect to various attributes. However as shown in Fig 7, these rules are able to sample tasks which have a lower accuracy than tasks which do not have any shift between the respective support and query sets. Moreover, designing such heuristic rules is not straightforward and requires a significant amount of human effort. FASTDIFFSEL effectively addresses this problem and provides a way to automatically extract difficult tasks, where the support and query set have shifts with respect to various factors.

## E MODEL CHOICES

In the past few years, a tremendous amount of research has been initiated in developing few-shot learning algorithms mostly spanning meta-learning (Snell et al., 2017; Lee et al., 2019; Bertinetto et al., 2019; Finn et al., 2017a). However recent works (Chen et al., 2020; Tian et al., 2020) have shown that a well-trained feature extracted via supervised learning is competitive and often surpasses state-of-the-art meta-learning algorithms. In (Hu et al., 2022), the authors show that upgrading the feature extractor and large-scale pre-training improves few-shot learning performance on challenging benchmarks such as MD. In our paper, we motivate our model choices from (Hu et al., 2022) and focus on a range of pretrained models which are adapted via Prototypical Networks (Snell et al., 2017) or full-model fine-tuning. Below we provide the architectural details for the models tested on HARD-MD++:

- **resnet18-101**. We use 4 different pre-trained ResNet (He et al., 2015) architectures (resnet-18,34,50,101) where the pre-training corpus is ilsvrc_2012 (Deng et al., 2009).
- **dino_small_patch16**. We use ViT-S/16 pre-trained with DINO (Caron et al., 2021) on ilsvrc_2012 (Deng et al., 2009).
- **dino_base_patch16**. ViT-B/16 pre-trained with DINO (Caron et al., 2021) on ilsvrc_2012 (Deng et al., 2009).

- **dino_base_21k**. ViT-B/16 pre-trained on ImageNet-21k (Deng et al., 2009).
- **vit_small_1k**. ViT-S/16 pre-trained on ilsvrc_2012 (Deng et al., 2009).
- **vit_base_1k**. ViT-B/16 pre-trained on ilsvrc_2012 (Deng et al., 2009).
- **beit_base_21k**. ViT-B/16 pre-trained using BeiT (Bao et al., 2021) on ImageNet-21k (Deng et al., 2009).
- **dino_resnet50**. ResNet-50 pre-trained with DINO (Caron et al., 2021) on ilsvrc_2012 (Deng et al., 2009).
- **clip_resnet50**. Visual encoder from CLIP (Radford et al., 2021) pretrained on YFCC100M - ResNet-50.
- **clip_vit_base**. Visual encoder from CLIP (Radford et al., 2021) pretrained on YFCC100M - ViT-B/16.

We believe that these model choices cover a wide-range of pre-training paradigms including both self-supervised and supervised pre-training.

## F  FINETUNING DETAILS

**Fine-tuning during adaptation.** We adopt the fine-tuning recipe from (Hu et al., 2022) for fine-tuning the pre-trained checkpoints. We specifically fine-tune the entire backbone for all the models, rather than partial adaptation. The fine-tuning algorithm from (Hu et al., 2022) has three hyper-parameters: (i) learning rate; (ii) number of fine-tuning steps and (iii) probability of switching on data-augmentation for the support set. In our experiments, we set the number of fine-tuning steps as 50 and the probability of switching on data-augmentation for the support set as 0.9. We select the optimal learning rate from $\{0.0001, 0.001, 0.01\}$ with the help of a separate validation set which comprises of 5 tasks per sub-dataset in HARD-MD++.

**ProtoNets during adaptation.** In this adaptation strategy, the support set is first used to compute class-specific prototypes. Then each query example is assigned the class, based on its closest distance to the class-prototypes. During adaptation, no parameters are learnt which makes it extremely fast during adaptation.

For both the types of adaptation on MD and HARD-MD++, images are resized to 128x128 following (Hu et al., 2022) to ensure fairness in comparison. We note that increasing the resolution to 224x224 can slightly improve performance on MD (Bronskill et al., 2021) with LITE training strategies however we leave it for future exploration.

## G  META-TRAINING DETAILS

The model used for extracting the difficult support set is meta-trained using Prototypical Networks (Snell et al., 2017) from its pre-trained initialization on the ilsvrc_2012 split from MD. We train this model using distributed training on 8 a6000 GPUs. We run the training for 50 epochs, where in each epoch 2000 episodes from ilsvrc_2012 are sampled. In total, we meta-train on 100k episodes variable-way, variable-shot episodes from ilsvrc_2012. We also train the models for 100 epochs, but observe that the best model gets selected before the $50^{th}$ epoch, hence resort to training for 50 epochs across. We train using the SGD optimizer with a momentum of 0.9. We use a learning rate of 5e-4 with cosine scheduler for our experiments. During training, following (Hu et al., 2022), we resize the images to 128x128.

## H  MORE STRESS-TESTING RESULTS

### H.1  FIXED-WAY, FIXED-SHOT RESULTS

Using our algorithm we also extract fixed 5-way, 5-shot tasks from MD. We evaluate all the 13 models on these fixed tasks from the original MD as well as HARD-MD. Overall, across both the adaptation strategies we find a drop of $\sim 30\%$ in performance on HARD-MD. Similar to the results for variable-way, variable-shot sampling, we find that ViT-B with CLIP initialization has a strong few-shot performance. In particular, ViT-B with CLIP has the best performance on HARD-MD when it is fully finetuned during adaptation, while ranks 2nd when Prototypical Networks is used during

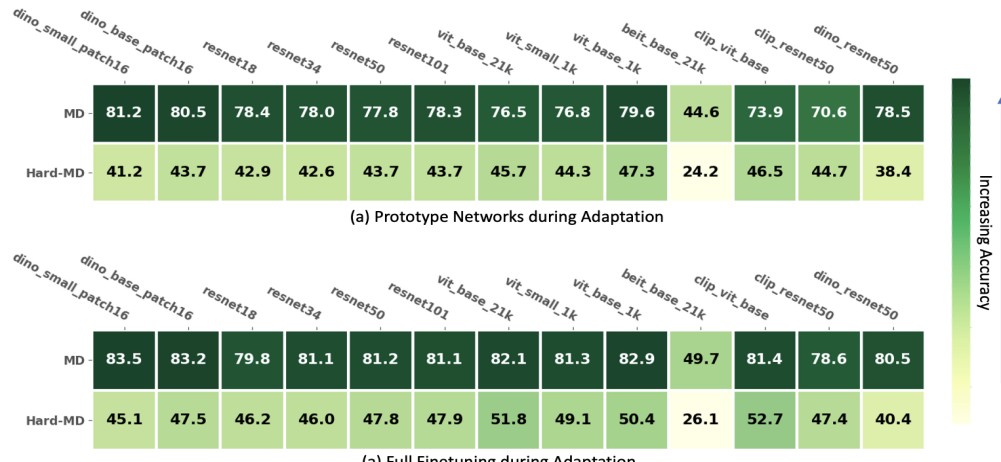

Figure 9: **Evaluation of fixed 5-way,5-shot tasks on META-DATASET and HARD-META-DATASET):** We find a consistent drop in performance across a fixed-way, fixed-shot setting with ViT-B(CLIP) performing the best with fine-tuning, while it ranks 2nd with Prototypical Networks.

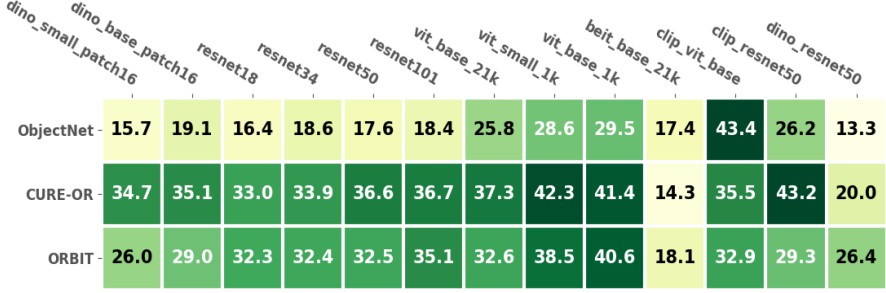

Figure 10: **Evaluation of fixed 5-way, 5-shot tasks on from OBJECTNET, CURE-OR and ORBIT with full-finetuning**: With ORBIT, we find that vision transformers (ViT-S and ViT-B) pre-trained on ilsvrc_2012 to be performing the best. For OBJECTNET and CURE-OR, we find that CLIP pre-training to be still performing the best amongst all the other models.

adaptation. However note that ViT-B with CLIP lags behind other models on fixed-way, fixed-shot tasks sampled from original MD.

The combined results with Section 4 imply that vision-language pre-training can offer more robustness to difficult tasks, than tasks sampled randomly. A thorough and more fine-grained study of vision-language models on HARD-MD can be a direction of future work.

## H.2 FINETUNING RESULTS FOR ORBIT, CURE-OR AND OBJECTNET

The full finetuning results for difficult tasks from OBJECTNET, CURE-OR and ORBIT can be referred at Fig 10. In particular for OBJECTNET, we find ViT-B with CLIP to be performing the best, while for CURE-OR, we find ResNet-50 with CLIP initialization to be performing the best. For ORBIT though, we find that vision transformers (ViT-S or ViT-B) pre-trained on ilsvrc_2012 to have the strongest performance.

## H.3 BREAKDOWN OF RESULTS FOR HARD-META-DATASET

In this section, we provide breakdown of the results on MD and HARD-MD at the sub-dataset level for both the adaptation strategies. In particular we report the average task accuracy along with the 95% confidence interval.

H.3.1    ADAPTATION: PROTOTYPICAL NETWORKS

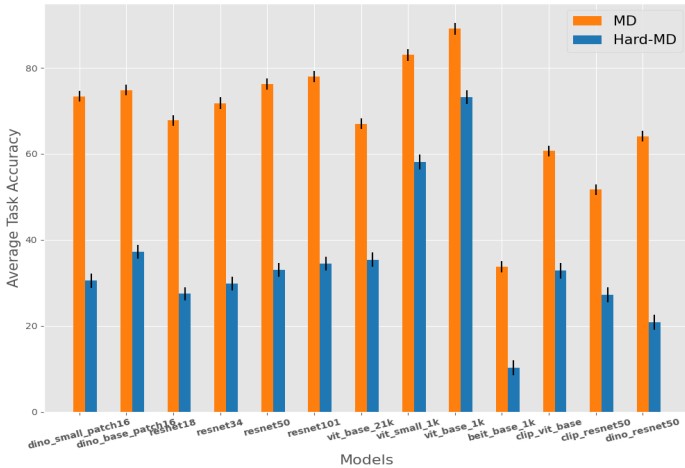

Figure 11: **ilsvrc_2012**

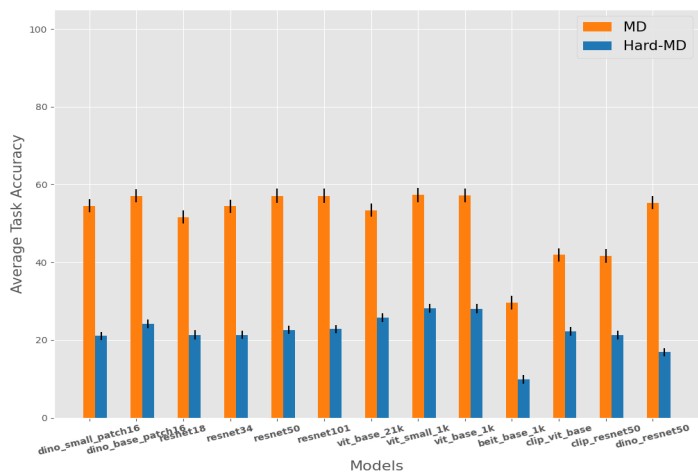

Figure 12: **MSCOCO**

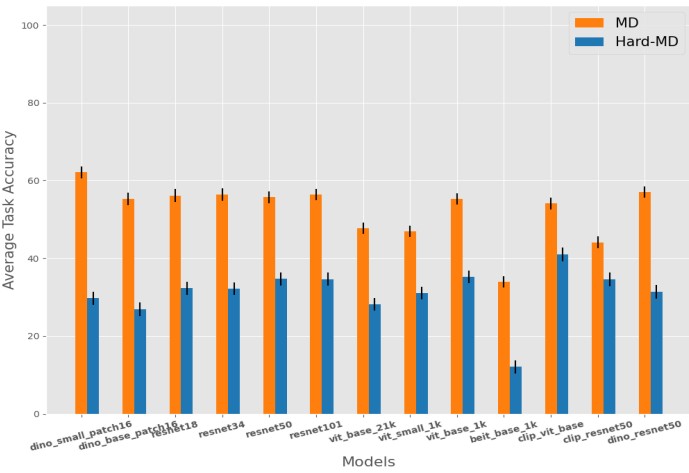

Figure 13: **Aircraft**

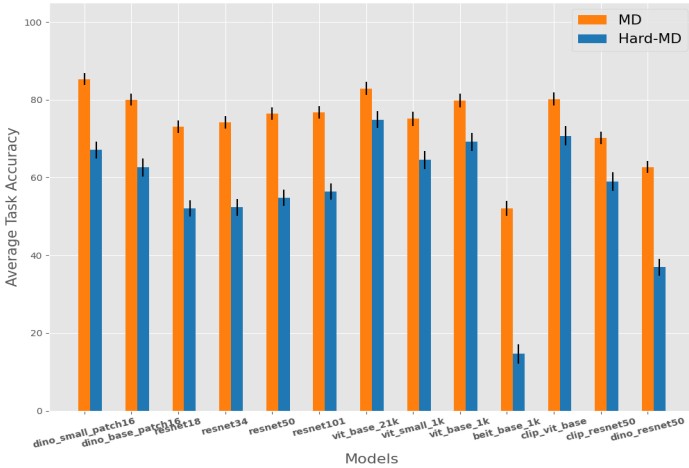

Figure 14: **Cu-birds**

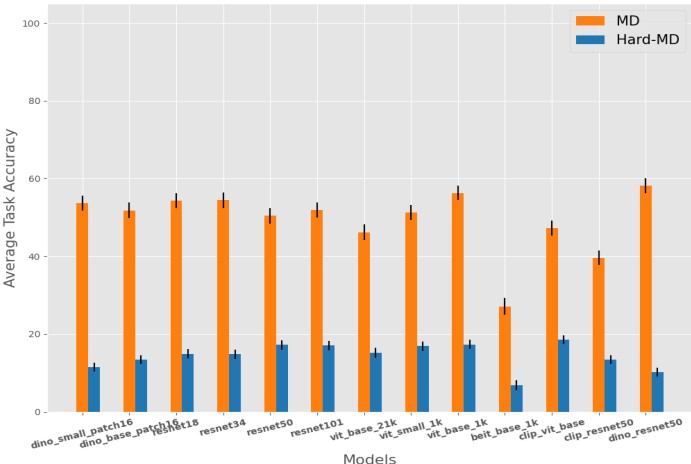

Figure 15: **Traffic-Sign**

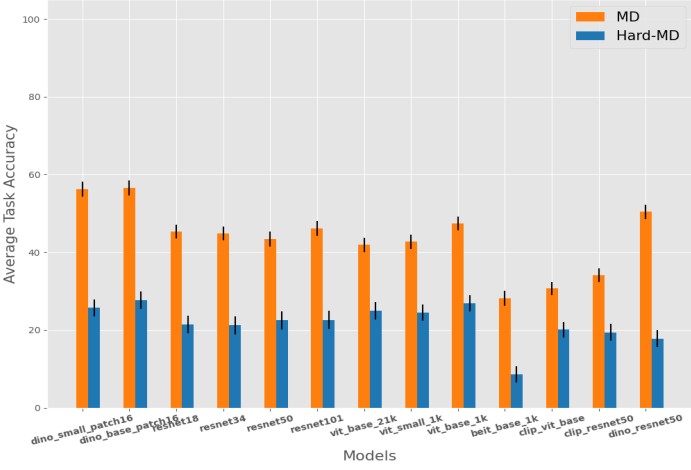

22

Figure 16: **Fungi**

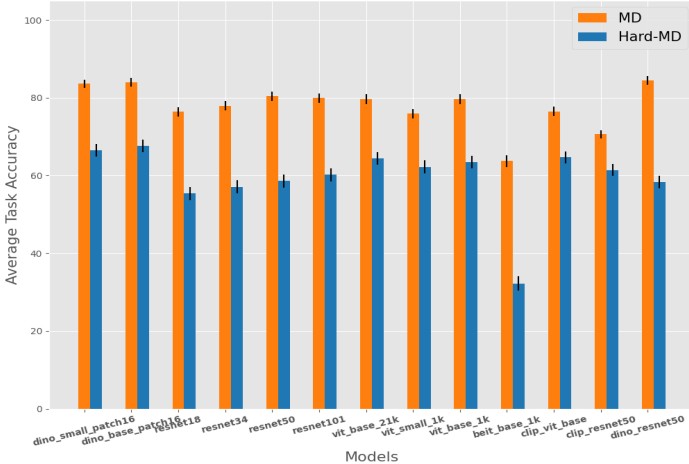

Figure 17: **DTD**

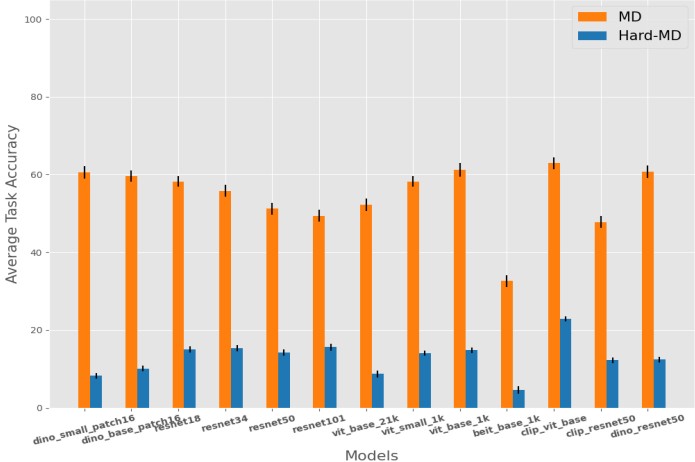

Figure 18: **Quickdraw**

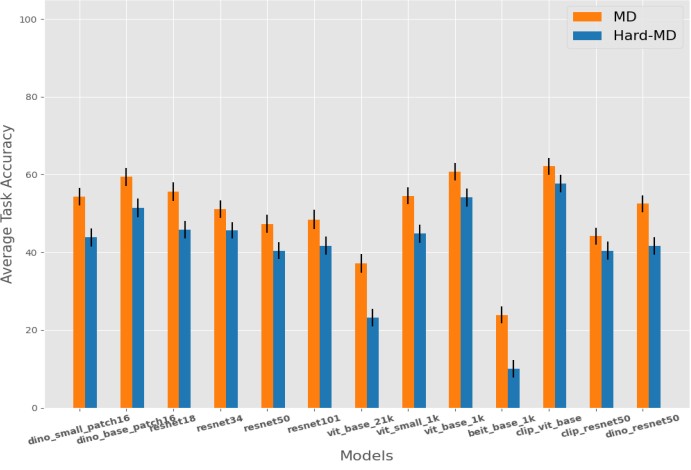

23

Figure 19: **Omniglot**

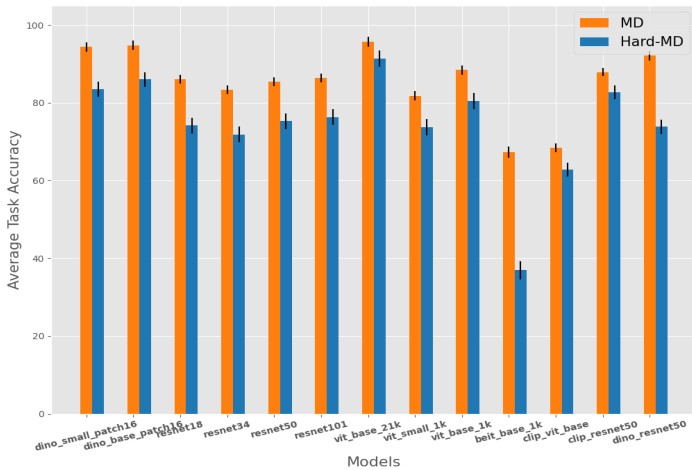

Figure 20: **VGG-Flower**

## H.3.2 ADAPTATION: FULL FINETUNING

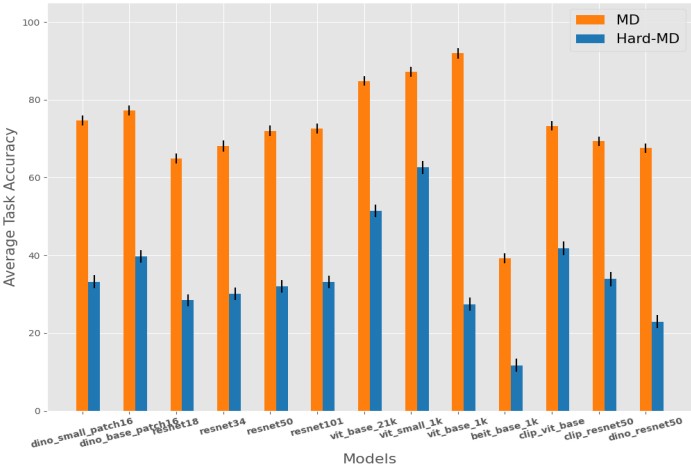

Figure 21: **ilsvrc_2012**

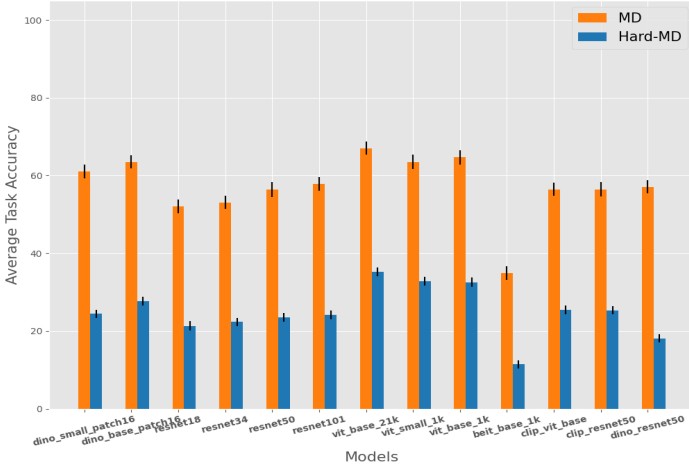

Figure 22: **MSCOCO**

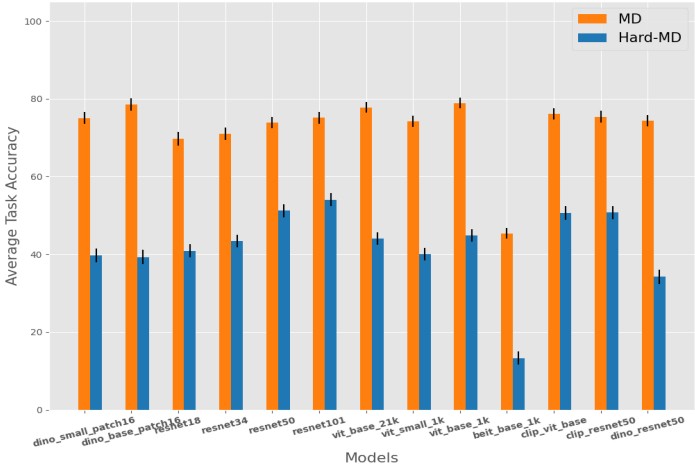

Figure 23: **Aircraft**

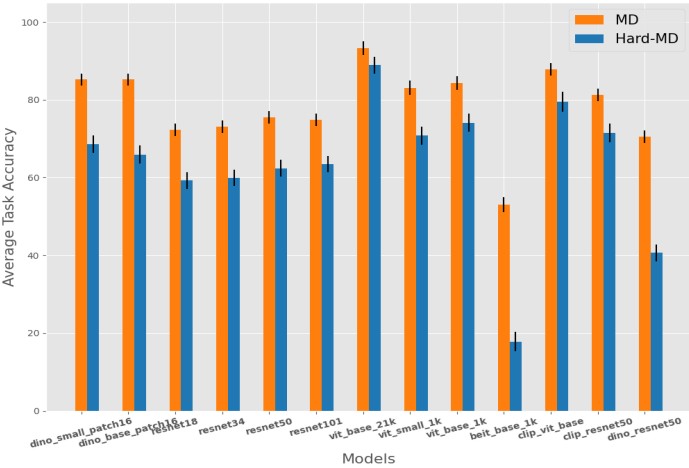

Figure 24: **Cu-birds**

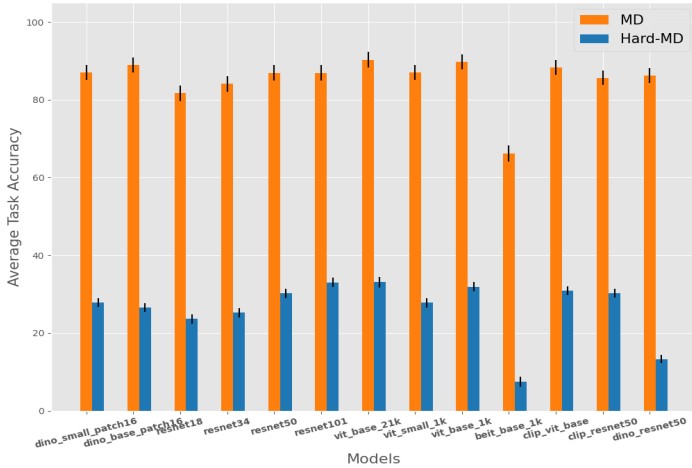

Figure 25: **Traffic-Signs**

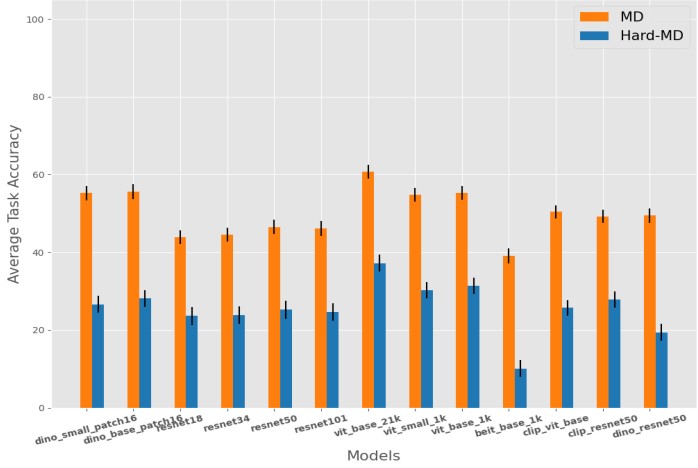

Figure 26: **Fungi**

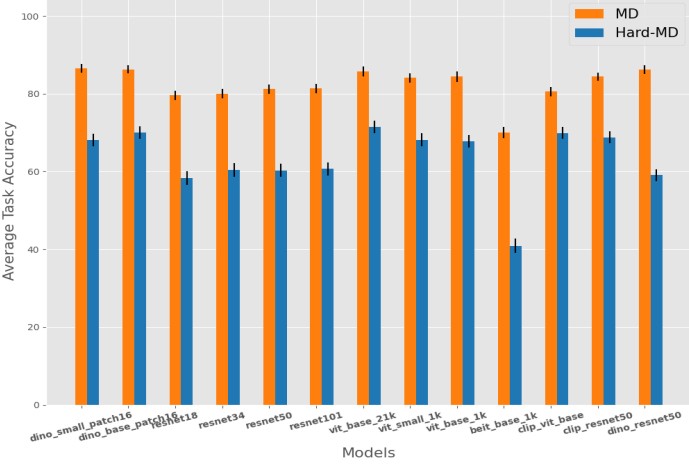

Figure 27: **DTD**

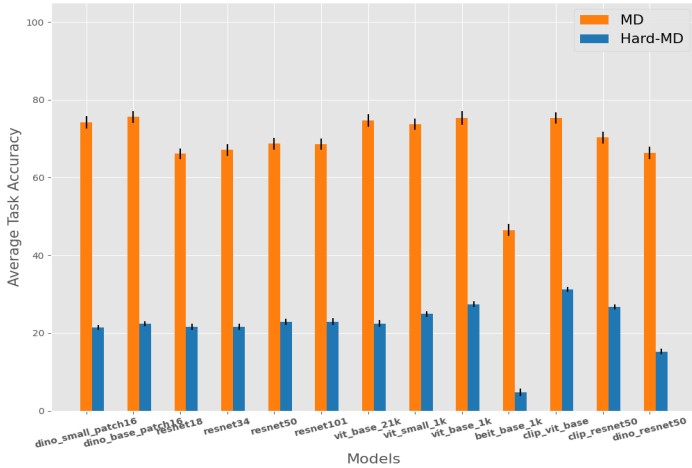

Figure 28: **Quickdraw**

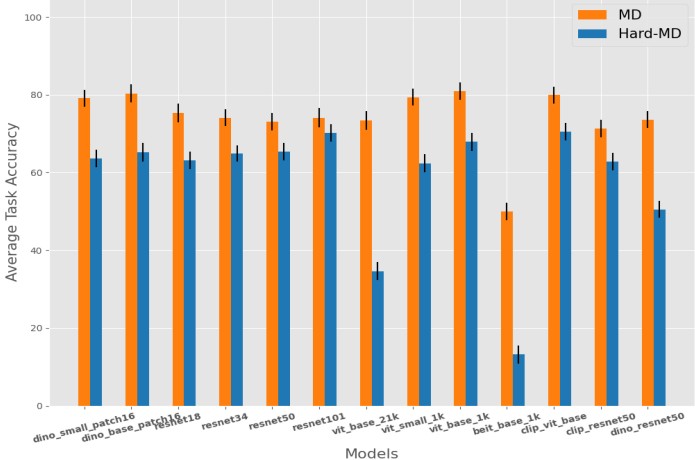

Figure 29: **Omniglot**

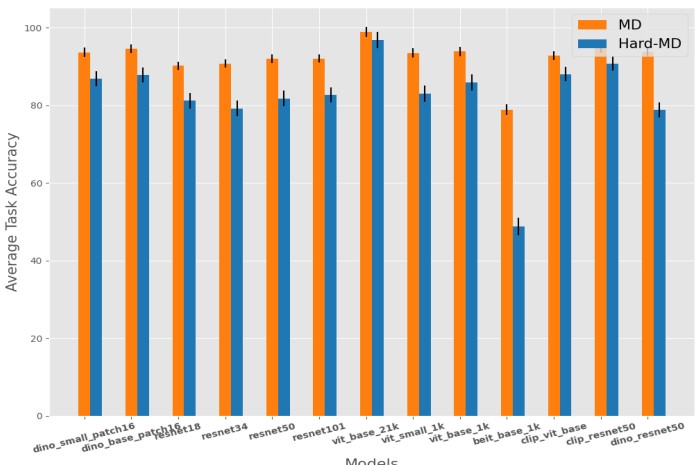

Figure 30: **VGG-Flower**

## H.4 IMPACT OF META-TRAINING ON ILSVRC_2012

We meta-train a subset of the 13 models on the ilsvrc_2012 split from META-DATASET. In particular, we meta-train dino_small_patch16, dino_base_patch16, dino_resnet50 and vit_base_1k from their pre-trained initialization. Prior work (Hu et al., 2022; Chowdhury et al., 2021) has shown that fine-tuning from a strong pre-trained checkpoint during adaptation is more beneficial than meta-training on ilsvrc_2012, therefore we primarily focus on the full fine-tuning experiments (see Section 4) in our paper. However, to verify if meta-training on ilsvrc_2012 leads to any significant improvements, we evaluate 4 meta-trained models on HARD-MD. From Fig 31, we find that the ilsvrc_2012 meta-trained models still suffer on HARD-MD and incur a ∼20-30% performance drop.

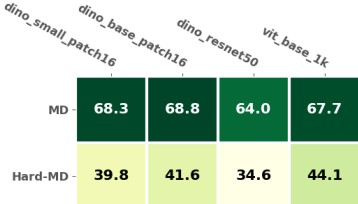

Figure 31: **Effectiveness of Meta-Training on the ilsvrc_2012 split from META-DATASET.** We observe a ∼20-30% drop in performance on the MD split from HARD-MD.

## I VISUALIZATION OF DIFFICULT TASKS

In this section, we provide visualizations of some of the difficult tasks which are extracted by our algorithm. We provide only a few visualizations for representative purposes and believe a large-scale qualitative study of the difficult tasks in HARD-META-DATASET++ is a future course of study.

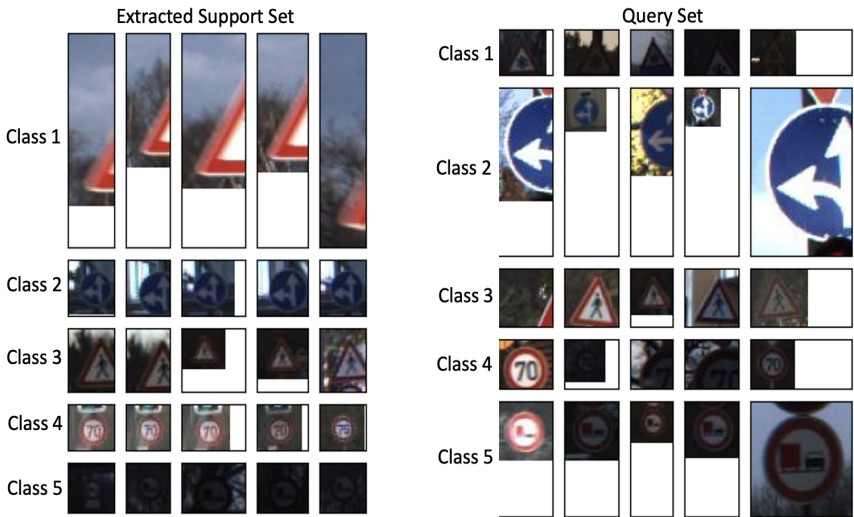

**Observation**: Change in lighting condition between support and query set for certain classes

Figure 32: **Difficult Task from Traffic-Sign:** This extracted task from traffic_sign has an accuracy of 15%. We find that this task has a distribution shift between the support and query with respect to various lighting conditions. For e.g., Class 5 in the extracted support set has almost no light when compared to the images from Class 5 in the query set.

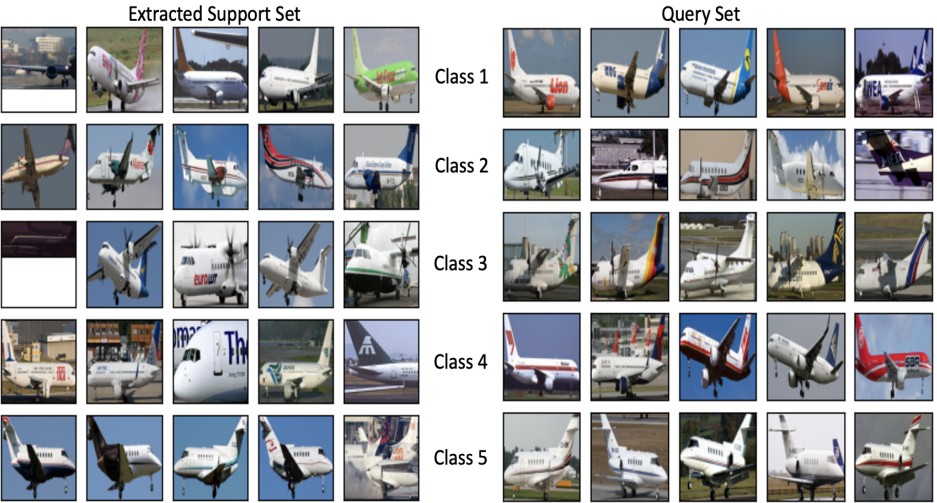

Figure 33: **Difficult Task from aircraft:** This extracted task from aircraft has an accuracy of 21%. We find that this task has a distribution shift between the support and query with respect to background. For e.g., Class 5 in the extracted support set has all the airplanes in the sky, whereas class 5 in the query set has all the airplanes on the ground. Similar for Class 1 and Class 2, where there is a distribution shift with respect to background characteristics.

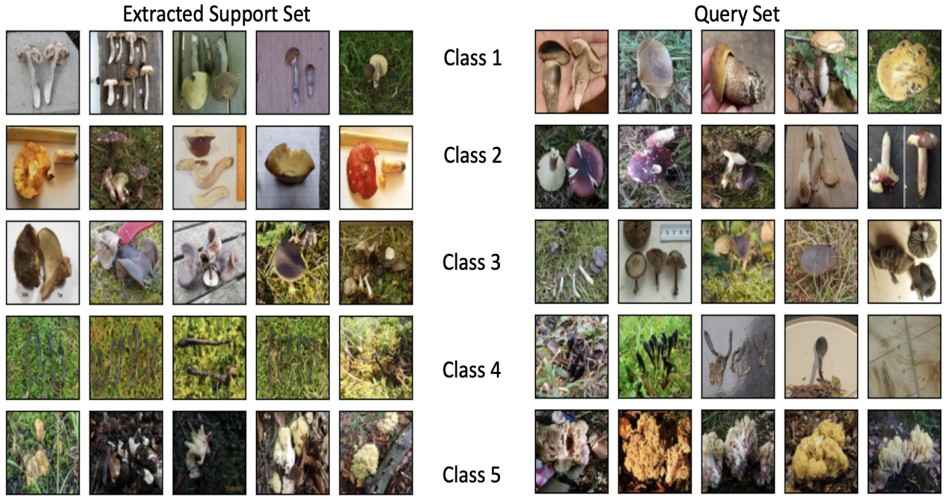

Figure 34: **Difficult Task from fungi:** This extracted task from fungi has an accuracy of 23%. For Class 4, our algorithm extracts images which consistently have the fungi in a background of vegetation, whereas in the query set *only* one image in Class 4 has a background of vegetation.

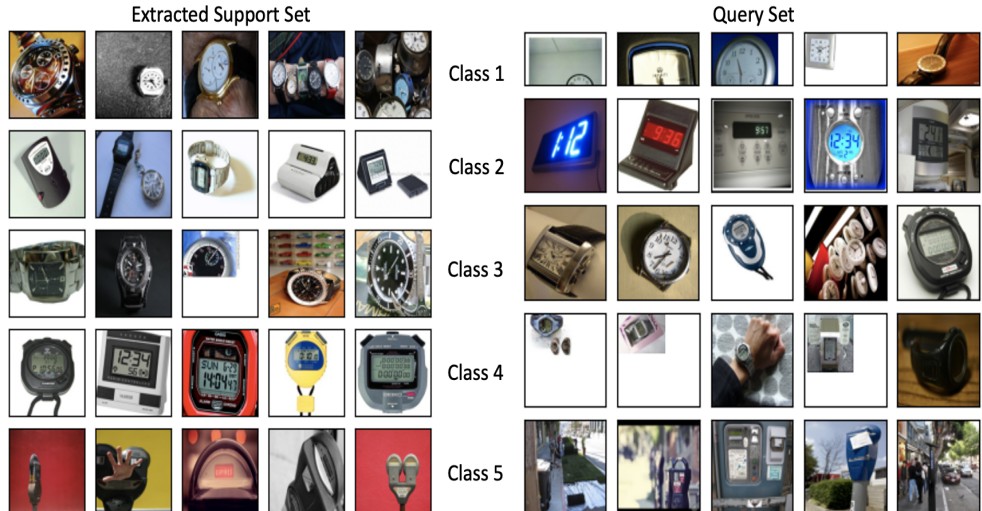

Figure 35: **Difficult Task from ilsvrc_2012:** Although we do not include ilsvrc_2012 in HARD-MD++, we present a difficult task which has an accuracy of 20%. Class 5 has a clear distribution shift with respect to viewpoint and background. There are multiple objects in the extracted support set of Class 1, whereas Class 1 in the query set only contains images of single objects.

These are some randomly picked tasks from the pool of difficult tasks extracted by our algorithm, where we find that difficult tasks have a distribution shift between the support and query with respect to various characteristics such as lighting, background, viewpoint. Annotation of the difficult tasks in HARD-MD++ is the next step to understand failure modes of few-shot learning methods on a deeper level.

## J  INTEGRATING OUR FRAMEWORK WITH FINE-GRAINED CLASS SAMPLERS

Our framework does not put any explicit constraints on the class-sampling procedure. As shown in Equation (1), our framework extracts difficult tasks where there is a weak relationship between the support and query. In all our experiments, we follow the same class sampling protocol from MD (Triantafillou et al., 2019). Note that the ilsvrc_2012 split from MD uses a fine-grained class-sampling protocol and our framework extracts more difficult tasks than the ones sampled using fine-grained class samplers for this setting (see Fig 30). Therefore, our framework is complementary to *any* fine-grained class-samplers. Recently (Bennequin et al., 2022) introduce a new fine-grained class-sampler for tieredImageNet consisting of tasks with classes which are semantically closer. The sampler can select tasks with different levels of coarsity, thereby providing control over the difficulty of the meta-test set. To understand the complementary nature of our framework, we integrate the class-sampling protocol from (Bennequin et al., 2022) with our optimization steps. We pre-train a ResNet-12 on the training split from tieredImageNet consisting of 351 classes following the superised pre-training protocol in (Bennequin et al., 2022). We then use this base model along with our framework to extract difficult tasks from the test-set of tiered-ImageNet.

|                            | 1st. Qtl | 2nd. Qtl | 3rd. Qtl | 4th. Qtl |
|----------------------------|----------|----------|----------|----------|
| Fine-grained protocol      | 42.01    | 51.2     | 56.7     | 62.7     |
| Ours + Fine-grained protocol | **24.3** | **29.3** | **31.4** | **33.1** |

Table 1: **Our framework is complementary to the fine-grained sampling protocol from (Bennequin et al., 2022)**. We find that our framework when used with the fine-grained sampling protocol extracts more difficult tasks, than fine-grained tasks itself, highlighting its complementary nature. Across 5000 tasks (5-way, 5-shot). Different quartiles represent different levels of task coarsity (smaller the coarsity, higher is the difficulty).

## K    EFFECT OF THE NUMBER OF OPTIMIZATION STEPS

In our framework, we take one step of gradient ascent and one subsequent step of the projection step per class. Empirically, we find that *only* one step is sufficient to extract difficult support sets, if the learning rate ($\alpha$) for gradient ascent is set as high as 200. In practice, in our experiment test-bed, we observe that the difficulty of the task decreases if the number of iterations increases to greater than 1. This could be attributed to the fact that the selection weight vector becomes sparse after one iteration, which might not be a good starting point for further steps of the optimization. Below we plot the effect of the number of iterations on the weighted prototype loss for a subset of the domains from HARD-MD++, which we use as a measure of the stopping criterion for our algorithm.

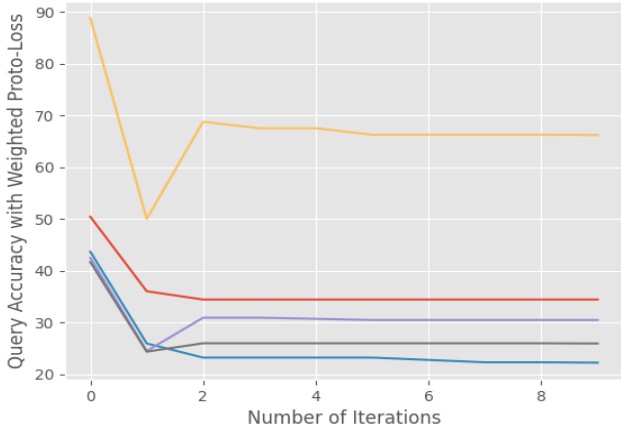

Figure 36: **Number of Optimization Steps (Omniglot).**

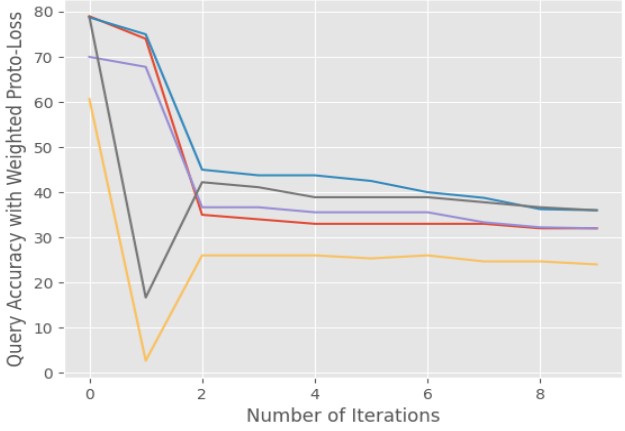

Figure 37: **Number of Optimization Steps (Aircraft).**

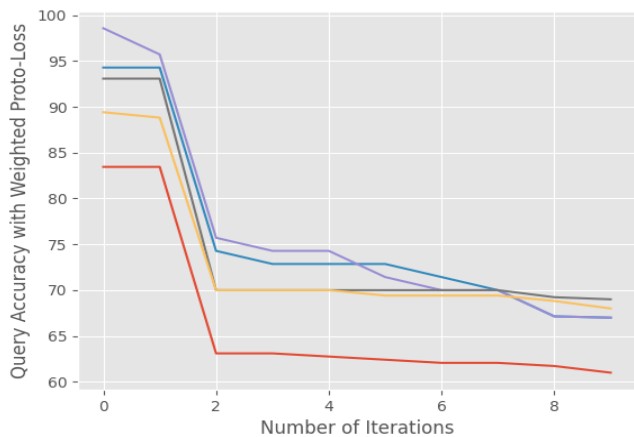

Figure 38: **Number of Optimization Steps (cu-birds).**

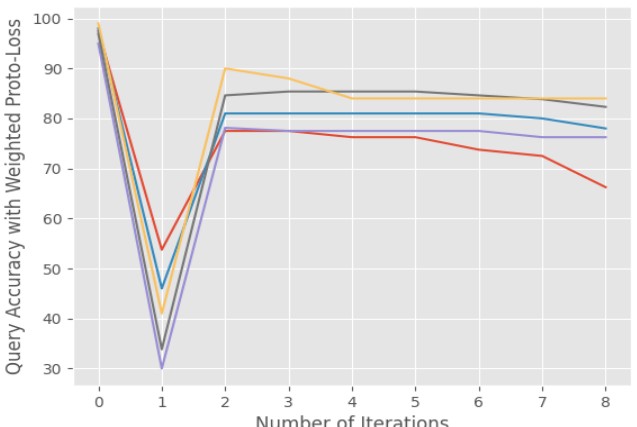

Figure 39: **Number of Optimization Steps (vgg-flower).**

Note that we use a post-processing step from the learnt selection weights to extract tasks with a given value of shot per class (i.e. $k_j$ samples for each of the $j^{th}$ class). Empirically, we track the query accuracy with the extracted support set and always find the lowest accuracy on the query set to be after one iteration of the algorithm.

## L  RESULTS ON MS-COCO (FULL)

The MS-COCO split in META-DATASET consists of object-cropped images based on the bounding box annotations. However, the original MS-COCO (Lin et al., 2014) (hereby denoted as MS-COCO (Full)) with bounding box and segmentation annotations can be a measurement for the number of objects present in the image. To understand, if a distribution shift between the support and query with respect to the number of objects is a factor in increasing task difficulty, we perform a fine-grained study using our framework and MS-COCO (Full). In particular we fix the query set to have images with objects in the set: $\{1, 2, 4, 6, 10\}$. For these fixed query sets, we use our framework to extract easy and difficult support sets.

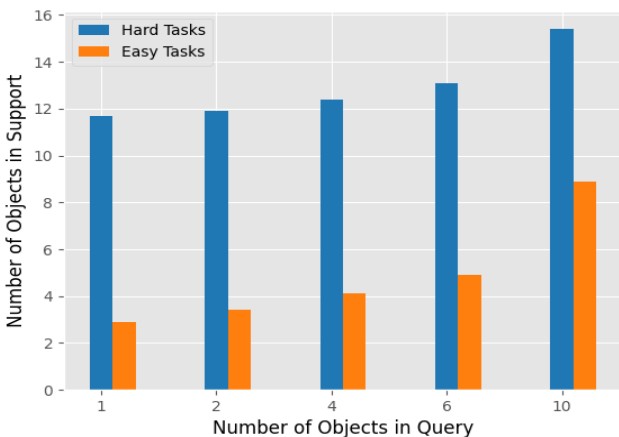

Figure 40: **Large distribution shift with respect to number of objects between support and query for difficult tasks from MS-COCO (Full).**

From Fig 40, we find that our framework extracts difficult tasks where there is a large distribution shift between the support and query with respect to the number of objects. Along with the results for ORBIT and CURE-OR (Section 5.3.2), this validates that our framework extracts tasks which have a weak relationship between the natural characteristics of the support and query set (e.g., occlusion, viewpoint, number of objects).

## M    RESULTS WITH 600 TASKS PER DATASET

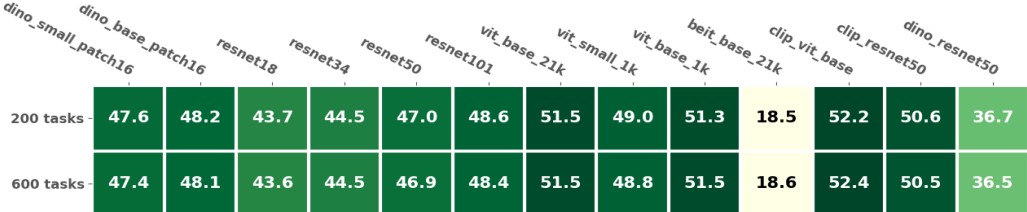

Figure 41: **Results with 600 tasks on HARD-MD.**

From Fig 41, we find that the ordering of the different models in terms of performance remain the same even when evaluated on 600 tasks per dataset. For e.g., ViT-B with CLIP initialization still has the best performance on the HARD-MD split from HARD-MD++.

## N    FORMULATION AND RESULTS WITH TASK2VEC

Task2Vec (Achille et al., 2019) is a task embedding method which can provide vectorial representations for tasks. The core idea in task2vec is to compute an embedding based on the diagonal of the Fisher information matrix of a probe network. Given a probe network $p_w$ parameterized by weights $w$, the task embedding is the diagonal of the Fisher Information Matrix (FIM) $F$:

$$F = \mathbb{E}_{x,y \sim \hat{p}(x)p_w(y|x)}[\nabla_w \log(p_w(y|x))\nabla_w \log(p_w(y|x)^T)] \tag{10}$$

In practice, the diagonal of the Fisher Information Matrix, which is a vector (where $t = Diag(F)$) is used as the task embedding. In principle task2vec can be used with our optimization framework to extract difficult tasks. Given a fixed query set $Q$ and a search pool $P$, where each example in $P$ has a selection weight $w_i$ associated to it:

$$\max_{\mathbf{w}} \ell(t(Q), t(P, \mathbf{w})) \tag{11}$$

|  | mscoco | traffic_sign | ilsvrc_2012 | omniglot | aircraft | cu_birds | dtd | quickdraw | fungi | vgg_flower |
|---|---|---|---|---|---|---|---|---|---|---|
| DINO + Proto-Loss | **21.06** | **11.4** | **30.5** | **43.8** | **29.7** | **67.1** | **66.5** | **8.2** | **25.7** | **83.5** |
| DINO + Task2Vec | 43.1 | 39.2 | 51.3 | 49.2 | 43.2 | 78.1 | 76.4 | 32.1 | 39.2 | 89.3 |

Table 2: **Accuracy of extracted tasks using Proto-Loss vs. Task2Vec across different domains from META-DATASET:** Using task2vec within our framework results in easier tasks than using Proto-Loss (current loss in our framework). Across 600 variable-way, variable-shot tasks per dataset. (Lower is better)

$$s.t. \quad w^i \in \{0, 1\}, \qquad \forall i \in [1, M]$$
$$\|\mathbf{w}_j\|_0 \le k_j, \qquad \forall j \in [1, N]$$

where $t$ is the task embedding from task2vec and $\ell(.)$ is the mean-squared error. We solve eq. (11) using the same projection algorithm used in eq. (3). In this framework, we use ViT-S pre-trained with DINO (Caron et al., 2021) as the probe network.

**Findings.** We find that using the task2vec formulation results in extracted tasks (Table. (2)) which are easier than the Proto-Loss formulation used in our framework eq. (1). Second, using task2vec is more computationally expensive than our method, as it involves computing the diagonal of the FIM which involves computing the gradients, whereas our current formulation only needs to compute the Proto-Loss which requires one forward pass.

## O    ABLATIONS WITH DIFFERENT BASE MODELS

|  | mscoco | traffic_sign | ilsvrc_2012 | omniglot | aircraft | cu_birds | dtd | quickdraw | fungi | vgg_flower |
|---|---|---|---|---|---|---|---|---|---|---|
| ViT (S) + DINO | 21.06 | 11.4 | 30.5 | 43.8 | 29.7 | 67.1 | 66.5 | 8.2 | 25.7 | 83.5 |
| ViT (B) + DINO | 21.4 | 11.9 | 31.3 | 43.2 | 28.8 | 67.9 | 67.1 | 9.1 | 25.3 | 83.9 |
| ViT (B) + CLIP | 22.4 | 13.4 | 35.4 | 46.7 | 29.1 | 70.1 | 69.1 | 13.1 | 27.1 | 84.1 |
| ViT (S) + DeiT | 22.1 | 14.2 | 34.1 | 45.1 | 30.1 | 69.3 | 70.1 | 11.3 | 26.5 | 83.4 |
| ViT (B) + DeiT | 22.3 | 12.3 | 33.2 | 43.9 | 31.3 | 71.1 | 68.1 | 13.2 | 28.1 | 83.6 |

Table 3: **Effectiveness of our framework in extracting difficult support sets using different base models**. Across 600 variable-way, variable-shot tasks per domain from META-DATASET.

In Table. (3), we find that our framework is flexible enough to be used with different base models for extracting difficult tasks. While in principle any base model can be used with our optimization framework, we choose ViT(S)+DINO as its the few-shot pipeline used by (Hu et al., 2022) which results in strong cross-domain few-shot performances on MD.

## P    INTEGRATION WITH DIFFERENT ADAPTATION STRATEGIES

In our paper, we choose the loss with prototypical networks (Snell et al., 2017) as the adaptation strategy due to it's fast adaptation. In principle, other relatively fast adaptation strategies such as R2D2 (Bertinetto et al., 2019) can be used. However, there is an inherent problem using adaptation strategies such as R2D2 with MD. The adaptation step requires using the entire search pool $P$ with the adaptation strategy. With META-DATASET's variable-way, variable-shot sampling protocol, the search pool $P$ can be extremely large due to a large number of ways in certain tasks. For e.g., R2D2 requires solving a ridge-regressor in the adaptation step, where the optimal ridge regressor weights are denoted by:

$$W = X^T(XX^T + \lambda I)^{-1}Y \tag{12}$$

where $X$ contains the embeddings of the support set and $X \in \mathbb{R}^{|P| \times e}$, where $|P|$ is the size of the search pool and $e$ is the embedding size of the features from the base model. $XX^T \in \mathbb{R}^{|P| \times |P|}$ and grows quadratically with the size of the search pool size, which can be as large as 50k for certain datasets in MD such as quickdraw or ilsvrc_2012. Given the high dimensionality of $XX^T$, its inversion will be extremely expensive making it infeasible to use it for fast difficult task extraction from large-scale vision datasets.

## Q    COMPARISON WITH FILTERED OUT DIFFICULT TASKS

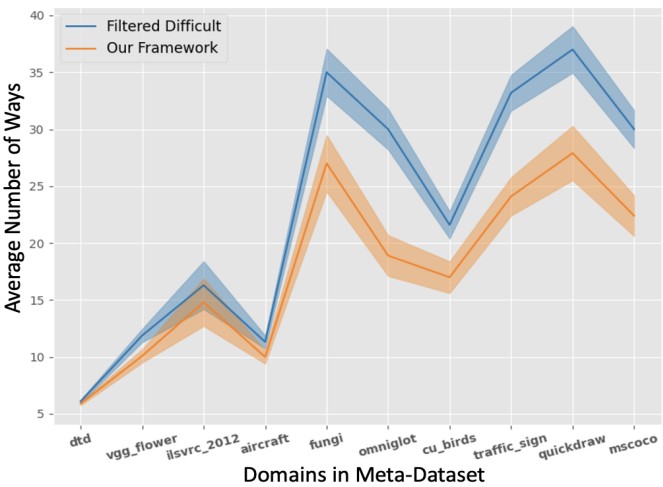

Figure 42: **Our extracted difficult tasks have lower number of ways than filtered out difficult tasks from META-DATASET's random sampler**.

We use our framework to extract 200 difficult tasks and run MD's random sampler for 10000 iterations and filter out 200 tasks with the lowest accuracies. We then compare the number of ways in both the set of difficult tasks. Previously (Triantafillou et al., 2019) has shown that a large number of ways in a given task can be a source of task difficulty. From Fig 42, we observe that across all the domains in MD, our framework is able to extract tasks with a lower number of ways than the filtered out difficult tasks from MD's random sampler. This shows that a large number of ways is not the primary source of difficulty in the tasks extracted by our framework. Furthermore in Section 5.3.2, through the lens of annotated datasets, we validate that the difficult tasks extracted by our framework have a distribution shift between the support and query with respect to various natural characteristics.

## R    RESULTS ON ADVERSARIAL TRAINING WITH DIFFICULT TASKS

We note that adversarial training with difficult tasks have been previously investigated in (Agarwal et al., 2021). Similar to the observations in (Agarwal et al., 2021), we find that adversarial training does not have intended improvements during meta-testing. In particular, we meta-train ViT-S (pre-trained with DINO) with prototypical networks on 10000 difficult training tasks extracted using our framework and 10000 randomly sampled tasks. We find that adversarial meta-training using difficult tasks does not lead to improvements on HARD-META-DATASET. This shows that one needs to carefully design new training paradigms for improving on the difficult tasks from HARD-META-DATASET.

|  | mscoco | traffic_sign | ilsvrc_2012 | omniglot | aircraft | cu_birds | dtd | quickdraw | fungi | vgg_flower |
|---|---|---|---|---|---|---|---|---|---|---|
| ViT (S) + DINO | 21.06 | 11.4 | 30.5 | 43.8 | 29.7 | 67.1 | 66.5 | 8.2 | 25.7 | 83.5 |
| ViT (S) + DINO + Adversarial Training | 21.4 | 12.3 | 30.1 | 42.3 | 29.5 | 66.5 | 67.1 | 9.1 | 26.1 | 83.2 |

Table 4: **Adversarial training does not improve performance on the HARD-MD split from HARD-MD++**. Across 600 variable-way, variable-shot tasks from each domain.

We note that the results in Table 4 are initial results on adversarial meta-training and more carefully designed adversarial meta-training methods might be required to improve performance on HARD-MD++.

## S    INITIAL RESULTS WITH TRANSDUCTIVE APPROACHES

We use the transductive fine-tuning approach introduced in (Dhillon et al., 2019a) and evaluate it on the tasks from HARD-MD++. We note that in our paper, we primarily evaluate inductive approaches in line with recently developed methods for MD (Triantafillou et al., 2019; Hu et al., 2022; Li et al., 2021). Evaluating and understanding the true effectiveness of transductive approaches will require a separate line of work. However, in this section we provide some initial results on how baseline transductive fine-tuning approaches fare on the HARD-MD split from HARD-MD++. Essentially, we add a regularizer to the fine-tuning cross-entropy loss to seek outputs from the query examples with a peaked posterior (i.e. low Shannon Entropy) similar to (Dhillon et al., 2019a).

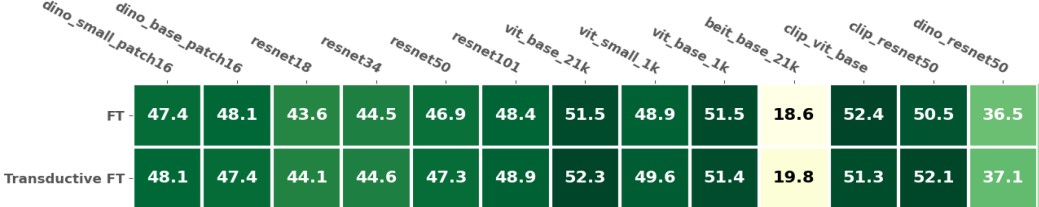

Figure 43: **Transductive Fine-tuning vs. Normal Fine-tuning (FT) on HARD-MD.**

From Fig 43, we find no clear benefit with transductive fine-tuning for the HARD-MD split from HARD-MD++. However, investigation of more transductive approaches such as (Boudiaf et al., 2020; Hu et al., 2020) is warranted to fully validate the effectiveness of transductive approaches and can be a good direction for future work.

## T    RESULTS WITH ADAPTERS

We leverage recently developed parameter efficient fine-tuning adapter based methods such as (Xu et al., 2022) and compare it with full network fine-tuning on the HARD-MD split from HARD-MD++.

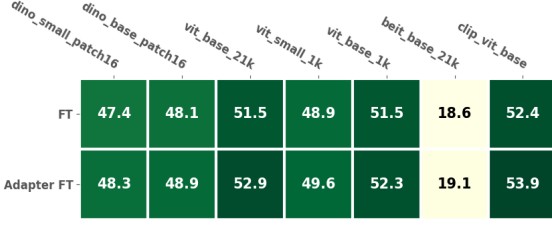

Figure 44: **Adapter Fine-tuning vs. Normal Fine-tuning (FT) on the HARD-MD split from HARD-MD++.**

Note that the adapter design in (Xu et al., 2022) is developed for ViT backbones and we find the effectiveness of their method on HARD-MD with 7 different vision transformer pre-trained backbones. In general from Fig 44, we find that carefully designed adapter based methods outperform full-network fine-tuning (though by a small margin). A thorough evaluation of parameter efficient fine-tuning techniques (Jia et al., 2022; Hu et al., 2021) on HARD-MD is a direction for future work.

