# OpenReview forum: "Hard-Meta-Dataset++: Towards Understanding Few-Shot Performance on Difficult Tasks"
_ICLR.cc/2023/Conference — ICLR 2023 poster_

### Official Review · Reviewer_tAM3 · 2022-10-24

**Confidence:** 5
**Correctness:** 2
**Technical Novelty And Significance:** 2
**Empirical Novelty And Significance:** 2
**Recommendation:** 3

**Clarity, Quality, Novelty And Reproducibility:**

The paper is easy to follow and well-structured. The proposed algorithm is well-detailed and pseudo-code is provided. The paper also gives enough information and details on the experimental part: design choices are motivated and all the hyper-parameters are given.

The novelty of the paper is low : it extends META-DATASET with hard tasks but the issues of the hard task in few-shot have been ever studied (Aggarwal et al, 2021). The paper just proposes a more efficient algorithm than this previous one to sample hard tasks.

**Strength And Weaknesses:**

**Strengths**

+ This paper targets an important issue in the few-shot classification community: the lack of benchmarks that are enough representative of real-world few-shot tasks. This problem is currently understudied. In particular, the proposed approach focuses on hard tasks defined as tasks for which a given few-shot classifier performs worse than for all other tasks (on average).
+ The paper is easy to follow and well-structured.
+ The resulting HARD-META-DATASET++ if it is publically released.

**Weaknesses**
+ My main concern is related to the proposed definition of a difficult task. The proposed definition is relative to other tasks but does not take into account the content of the task itself, i.e. the labels and the image. Indeed, in real-world scenarios, hard tasks are for instance fine-grained classification ones or tasks with shifts between the support sets and the query set.
+ Moreover, the proposed approach to sample hard tasks depends on the chosen loss, here proto-loss but also on a meta-trained based model. This point should be more discussed and in particular what is the impact of the choice of these two ingredients on the findings of the paper.
+ The paper title mentioned understanding few-shot performance but the "understanding part" is not enough detailed and developed in the paper.
+ Some important references are missing such that the paper of [(Benequin el al, 2022)](https://openaccess.thecvf.com/content/CVPR2022W/VDU/html/Bennequin_Few-Shot_Image_Classification_Benchmarks_Are_Too_Far_From_Reality_Build_CVPRW_2022_paper.html) (Few-Shot Image Classification Benchmarks are Too Far From Reality: Build Back Better with Semantic Task Sampling, that also tackles the weaknesses of current few shot benchmarks for real-world applications. Moreover, various approaches have been proposed to define a task such as for instance [Task2vec](https://arxiv.org/abs/1902.03545) (Achille et al, 2019) and the paper should mention them and position compared to them.
+  Transductive inference is widely used in few-shot learning. What about transductive few-shot learning in the proposed approach?



**Summary Of The Paper:**

This paper proposes a new algorithm to build efficiently a few-shot classification benchmark, named HARD-META-DATASET++ tailored to difficult tasks. In particular, the proposed algorithm is able to sample difficult tasks from various large-scale vision datasets in a deterministic way using a constrained optimization problem. The principle is to learn, through this constrained optimization problem, selection weights for each instance in a search pool in order to build a support set associated with a given query set. Compared to previous related approaches that use greedy search, the proposed algorithm is faster and thus enables to sample tasks in large-scale vision datasets. Using this algorithm, they build HARD-META-DATASET++ by sampling difficult tasks on META-DATASET, ORBIT, CURE-OR and OBJECT-NET. They then proposed to test various few-shot classification state-of-the-art approaches on HARD-META-DATASET++ and they show that, on average, these approaches fail on these hard tasks.

**Summary Of The Review:**

The paper proposes a constrained optimization algorithm for few-shot classification hard task sampling.  As such, I believe that the contributions of this paper are not significant enough to be published at the ICLR. In particular, as such the paper just brings a new benchmark on which state-of-the art approaches are weaker but it does not address at all the problem of better understanding and definition of why a task is difficult in few-shot, in particular in real-world settings. More insights are needed on this particular point and in relation with the images and labels of the targeted task.

---

> ### Author Response · Authors · 2022-11-13
> **Response to Reviewer tAM3**
>
>  **As such, I believe that the contributions of this paper are not significant enough to be published at the ICLR. In particular, as such the paper just brings a new benchmark on which state-of-the-art approaches / Novelty**:  The main technical novelty of our paper is a novel and efficient algorithm for extracting difficult tasks (>20x faster than existing methods). This is by no means a trivial contribution as without this fast extraction algorithm, one cannot study failure cases in few-shot learning and thereby cannot build a benchmark to stress-test few-shot models. We also want to emphasize that benchmarks are important to drive research forward (e.g., Meta-Dataset  [ICLR] or ORBIT [ICCV]).  Our benchmark provides a new evaluation suite for probing failure modes in few-shot learning which is an underlooked but critical area of research. In our paper, we also highlight significant empirical results (e.g., vision-language pre-training is beneficial for difficult tasks) and understanding of the extracted difficult tasks through the lens of annotated datasets. Together our proposed algorithm, Hard-MD++ and our analyses can be used to drive future algorithmic innovations and failure mode insights in few-shot learning.
>
>  **definition of why a task is difficult in few-shot, in particular in real-world settings; The proposed definition is relative to other tasks but does not take into account the content of the task itself, i.e. the labels and the image** :
> Thank you for bringing this point up. Our introduction’s definition of a hard task (“one for which a given few-shot classifier performs significantly worse on its query set compared to the mean query performance over all tasks”) is indeed misleading as does not reflect our actual formulation which instead aims to extract a support set which maximally reduces the accuracy on the query set (i.e. captures a distribution shift between support and query). We have updated our introduction to the following definition: “a difficult task is one on which a few-shot classifier performs poorly on the task's query set, after being adapted to its support set”.
>
> Although our formulation (and new definition) does not explicitly call out the content of the task or specific sources of difficulty, we can begin to get at these questions when we apply our algorithm to annotated datasets, as we show for ORBIT and CURE-OR in Sec. (5) and Appendix (J). We refer to the reviewer to comment on “but the "understanding part.."  and our “Analysis of Hard Tasks” response to Reviewer  trsF for further comments . We also note that our formulation/definition builds off on existing published works (e.g. Agarwal et.al (2021), Fu et.al (2022)) in this space.
>
>  **hard tasks are for instance fine-grained classification ones**: Even though our work does not explicitly focus on fine-grained classification tasks, our proposed algorithm is highly flexible and could easily be applied to fine-grained sampling protocols. For this rebuttal, we used our algorithm to extract difficult tasks from MD’s ilsvrc_2012 dataset, which contains fine-grained object categories. In Fig. (19), we see that models perform worse on our extracted fine-grained tasks, compared to fine-grained tasks from MD’s random sampler. We also integrated our proposed algorithm with the sampler from Benequin el al, 2022, as suggested by the reviewer, and have included details and new results in Appendix (H). Our results show that again, models perform worse on our extracted fine-grained tasks than tasks from Benequin el al, 2022’s fine-grained sampler.
>
> **or tasks with shifts between the support sets and the query set** :  We request the reviewer to refer to our “Analysis of Hard Tasks” response to Reviewer trsF where we validate that our proposed algorithm extracts tasks which have a distribution shift between the support and query, with respect to natural image characteristics.

---

> > ### Author Response · Authors · 2022-11-13
> > **Further Responses**
> >
> > **Impact of the choice of the adaptation function and the choice of the base model**:
> > We choose ProtoNets as the adaptation function primarily because it is fast and computationally efficient since it has no learnable parameters and hence does not require any optimization steps that another adaptation strategy (e.g. R2D2, linear-head) would for each step in our algorithm. This computational savings enables us to extract a large number of difficult tasks from large-scale vision datasets. In Appendix (N), we have added further analysis on why other adaptation strategies (e.g. R2D2) are not suitable to be used for difficult task extraction (e.g. due to expensive matrix inversion in R2D2).
> > We agree, however, that quantifying the impact of base model is also important so in Appendix (M) we have added new experiments using a large range of base model initializations and architectures for extracting difficult support sets (with ProtoNets as the adaptation) . Across different base model initializations (e.g., DeiT/CLIP/DINO), we find that our method is able to consistently extract difficult tasks.
> >
> > **Positioning with  (Benequin el al, 2022)**: Thank you for pointing us to this recent work where the class sampling procedure is changed to incorporate a degree of fine-grained-ness in tasks. Our proposed algorithm is complementary as it does not put any restrictions on class sampling and can easily be integrated with Benequin el al, 2022. In Appendix (H), we provide new results showing that our algorithm can flexibly be integrated with a fine-grained class-sampling procedure. Our results show that integrating our method with fine-grained sampling protocols results in the extraction of tasks with a lower accuracy than Benequin el al, 2022’s fine-grained sampler..
> >
> > **Positioning with respect to task2vec**: We thank the reviewer for pointing out task2vec. We did consider using task2vec with our optimization in the initial phase of our project, however we found that it led to inferior difficult task extraction (by ~15%) compared to our simple metric-based loss on Prototypical Networks. We have included these results in Appendix (L). We find that using task2vec within the optimization is ~10 times slower in extraction time (in practice).
> >
> > **Transductive inference is widely used in few-shot learning**:  We thank the reviewer for pointing this out. We have added a new strong baseline transductive method (Dhillon et. al(2020)) on HardMD++ to Appendix (Q). In general, we find that the transductive method consistently struggles on Hard-MD++, suggesting that our benchmark is indeed challenging for transductive and inductive few-shot methods alike. We highlight, however, that our paper primarily focuses on the inductive setting (similar to MD), where only one test example is available at one time during inference.
> >
> >  **but the "understanding part" is not enough detailed and developed in the paper**:  Our paper cannot hope to fully understand the robustness of few-shot classification to difficult tasks, but we do believe we’ve made a start ‘toward understanding’ these issues.
> >
> > We call out some our specific findings toward understanding few-shot robustness as well as point the reviewer to our “Analysis of Hard Tasks” response to Reviewer trsF :
> >
> > * A difficult task arises when there is a shift between the properties of its support and query set. We empirically validated this using 3 vision datasets (ORBIT, CURE-OR, and a third new dataset for this rebuttal, MS-COCO). This provides a more nuanced understanding of ‘why’ a task is difficult.
> > * Shift with respect to certain natural characteristics such as viewpoint, occlusion and number of objects are critical in making a task difficult.
> > * In particular, we find that vision-language pre-training (e.g. CLIP) performs well on difficult tasks (i.e. where there is a shift between support and query with respect to natural characteristics)

---

> > > ### Author Response · Authors · 2022-11-17
> > > **Further questions and comments**
> > >
> > > We thank the reviewer for the time and effort in reviewing our paper and providing constructive suggestions. This has certainly improved our paper. We hope that our response addresses your concerns and comments. Since we are reaching the end of the discussion period, if there are any further questions or comments we can address, please let us know!
> > >
> > > --Authors

---

> > > > ### Author Response · Authors · 2022-12-08
> > > > **Looking Forward to your Response!**
> > > >
> > > > Dear Reviewer  tAM3,
> > > >
> > > > We thank you again for your valuable feedback and comments which has helped to strengthen our paper. As the discussion period is ending soon, we would really appreciate if you could let us know if our responses have addressed your concerns. We will be happy to answer any further questions and address any remaining concerns.
> > > >
> > > > —Thanks
> > > >
> > > > Authors

---

### Official Review · Reviewer_yM4m · 2022-10-25

**Confidence:** 2
**Correctness:** 3
**Technical Novelty And Significance:** 3
**Empirical Novelty And Significance:** 3
**Recommendation:** 6

**Clarity, Quality, Novelty And Reproducibility:**

The paper is easy to read and follow.

Casting the hard episode selection as an optimization problem is interesting.

The algorithm should be easy to reproduce.

**Strength And Weaknesses:**

Strength
- casted the difficult support set extraction as a constrained optimization problem, to optimize the "selection weights" (eq 1.)
- solved the optimization problem using:
   1) ProtoNet as feature extractor
   2) applies 1-step gradient descent w.r.t the "selection weights"
   3) projection w and uses l1 relaxation, and solved the relaxed version using its dual form
- The difficult episodes are obtained by drawing the examples with the highest "selection weights"
- The algorithm can speedups of difficult episode search by 20x
- The constructed hard datasets is difficult for FSL models (fig 3)


Weakness
- Does number of steps have any effect on the final set? E.g., the current optimization of the "selection weights" uses 1 projection step.
- The optimization involves using f, and its set to a ProtoNet. Does the selection of f affect the accuracy of the difficult set selection?
- There are other methods to extract hard episodes, have the author compared with existing approaches?

**Summary Of The Paper:**

The authors proposed a novel efficient algorithm to extract hard tasks from existing large-scale datasets.

Finding the difficult episodes, (support set, query set) pairs, is a combinatorial problem, and the authors propose to use a model’s loss on the task’s query set, as a proxy for the difficulty of the task, and cast the problem as an optimization problem.

Using the proposed method, the authors extracted a dataset from (MetaDataset, ORBIT, CURE-OR and OBJECTNET).

The resulting dataset is used as a test-only few-shot classification benchmark, and is named HARD-META-DATASET++. The authors evaluate a few popular few-shot classification methods on this benchmark.

**Summary Of The Review:**

The paper presents an interesting way to construct the difficult subset from large scale datasets. The prospered methods shows promising effectiveness in construction such difficult set. The authors successfully constructed the HARD-META-DATASET++, and conducted the evaluation of various feature extractors (fig 4). However, the paper lacks comparison between the proposed approach and existing approaches. The proposed method also has some constraints: hand-picked f (ProtoNet), and the authors used 1 gradient step to solve the optimization wo/ justification.

---

> ### Author Response · Authors · 2022-11-13
> **Response to Reviewer yM4m**
>
> **Number of optimization steps**: In our experiments, we take one gradient ascent step and one projection step per class iteratively. We find that with a high learning rate, one step is sufficient to extract difficult few-shot tasks.. In Appendix (I), we provide further ablations on the number of steps required to extract difficult tasks.  One major advantage of using one optimization step is the large speed-up, which is an important factor when extracting a large number of difficult few-shot tasks from large-scale vision datasets.
>
> **Selection of the adaptation function f**: In our framework, we use Prototypical Networks because it is fast and computationally efficient since it has no learnable parameters and hence does not require any optimization steps that another adaptation strategy (e.g. R2D2 or a linear-head) would for each step of our algorithm. This computational savings enables us to extract a large number of difficult tasks at scale. Furthermore, we see that top-performing models across a wide range of method classes perform poorly on HardMD++, thereby showing the generalizability of these difficult tasks and empirically validating our choice of adaptation function. We would expect to see similar poor performance on tasks extracted with other adaptation functions (e.g. R2D2), but these would likely come at a higher computational cost (e.g., expensive matrix inversion for R2D2 even after Woodbury relaxation). We have added further information on the choice of this adaptation function and the drawbacks of other adaptation functions in Appendix (N).
>
> **Other methods to extract hard tasks**:  To the best of our knowledge, the greedy search-based algorithm proposed by Agarwal et.al (2021) is the only method which extracts difficult support sets from vision datasets for few-shot classification. Prior works from the subset selection literature [1, 2] (though not in the context of FSL), could potentially be adapted to our use-case, but would require a separate investigation in itself. We have cited these papers and added a short discussion to our manuscript’s related works.
>
> [1]. Glister: A generalization based data selection framework for efficient and robust learning: AAAI 2021
>
> [2]. Submodularity in Data Subset Selection and Active Learning: ICML 2015

---

> > ### Author Response · Authors · 2022-11-17
> > **Further questions and comments**
> >
> > We thank the reviewer for the time and effort in reviewing our paper and providing constructive suggestions. This has certainly improved our paper. We hope that our response addresses your concerns and comments. Since we are reaching the end of the discussion period, if there are any further questions or comments we can address, please let us know!
> >
> > --Authors

---

> > > ### Author Response · Authors · 2022-12-08
> > > **Looking Forward to your Response!**
> > >
> > > Dear Reviewer yM4m,
> > >
> > > We thank you again for your valuable feedback and comments which has helped to strengthen our paper. As the discussion period is ending soon, we would really appreciate if you could let us know if our responses have addressed your concerns. We will be happy to answer any further questions and address any remaining concerns.
> > >
> > > —Thanks
> > >
> > > Authors

---

### Official Review · Reviewer_trsF · 2022-10-25

**Confidence:** 5
**Correctness:** 4
**Technical Novelty And Significance:** 2
**Empirical Novelty And Significance:** 2
**Recommendation:** 6

**Clarity, Quality, Novelty And Reproducibility:**

The paper is well written, and of good quality. The originality of the work is good.

**Strength And Weaknesses:**

Strength:
1.	The idea of analyze the performance on hard tasks is practical.
2.	The proposed algorithm is simple yes effective.


Weaknesses:
1.	While the authors provide abundant results, the corresponding analysis is not so insightful. It is common sense that there are many potential reasons that can lead to low accuracy on query samples. For example, the support samples can either be hard to fit or easy to overfit, which are totally different when considering the properties of these hard tasks. It is hard to understand why these previous methods have large performance gap between the original MD and Hard-MD++ solely based on the average accuracy. It would be better if other metrics can be provided, e.g. the training loss curves on each dataset, the relation between the performance and shot/way number, etc.

2.	The paper would be more comprehensive if other adaptation methods like DCM [1], TSA [2] and eTT [3] can be used in the experiments.

3.	I wonder if it is possible to measure the average ‘hardness’ of the original experiment setting on MD, i.e. 600 episodes for each dataset. This is helpful for indicating the chance of handling such hard tasks in real-life scenarios where hard and easy tasks are mixed.

[1] Powering Finetuning for Few-shot Learning: Domain-Agnostic Bias Reduction with Selected Sampling. AAAI 2022
[2] Cross-domain few-shot learning with task-specific adapters. CVPR 2022
[3] Exploring Efficient Few-shot Adaptation for Vision Transformers. TMLR 2022


**Summary Of The Paper:**

This paper mainly focuses on few-shot learning. Different from the previous methods pursuing good overall accuracy, the authors study the performance on hard few-shot tasks. Specifically they propose a novel algorithm based on constrained optimization to find difficult support samples given query samples. Such a method is much efficient than greedy search. Based on the proposed method, the authors conduct extensive experiments on a new version of Meta-Dataset using existing methods and provide many interesting results.

**Summary Of The Review:**

This paper presents a new Few-shot learning setting, in order to better understand the difficulty of FSL.
My personal suggestion is that some insightful analysis may be appreciated, and importantly some recent adaptation methods may be compared in this setting. This can show that this setting is really meaningful.

---

> ### Author Response · Authors · 2022-11-13
> **Response to Reviewer trsF**
>
> **Results with TSA and eTT**:  We thank the reviewer for introducing us to these recent adaptation methods. We have added new results investigating eTT on Hard-MD++ in Appendix (R), where we see that these newer methods still suffer on Hard-MD++. This highlights that we need completely novel algorithmic innovations in order to tackle hard tasks, and this is something that Hard-MD++ can help to drive in the future.
>
> **Analysis of Hard Tasks**:
> [Real-World Image Qualities]: The 9 datasets from the original MD in Hard-MD++ are not annotated with any image characteristics, hence it is difficult to fully understand the cause of the performance drop. The question is nevertheless an important one, hence we leverage 2 annotated datasets in Hard-MD++: ORBIT and CURE-OR, to understand the properties of difficult tasks. In Section (5) in the main paper, we show that difficult tasks have a distribution shift in the characteristics between support and query. For e.g. in Fig. (5) for CURE-OR, we find that when the query examples have objects in the front or back viewpoint, the difficult tasks have support set with examples dominated by side viewpoints thereby designating a shift in characteristics.
>
> [Number of Objects - MSCOCO (Full)]: In addition, for this rebuttal we used our proposed algorithm to extract difficult tasks from MS-COCO - Full which is annotated with the number of objects in each image. Our results show that a task is difficult if the number of objects in an image greatly differs between a support and query set (see Appendix (J)). These results further support our paper’s main finding that a task is difficult if there is a distribution shift in the properties between its support and query set.
>
> [Number of ways / shots]: In Appendix (K), we also include new analyses which show that if we filter out difficult tasks from MD’s random sampler (i.e. extract tasks with low accuracy), these tasks are dominated by a high number of ways. This suggests that other sources of task difficulty might be masked out. In comparison, our proposed algorithm extracts difficult tasks with a lower number of ways, on average, highlighting that HardMD++ can be used to uncover new sources of task difficulty.
>
> **Measuring average hardness in original MD**:  This is an interesting idea and our proposed algorithm could be leveraged to achieve such a metric. One could train a surrogate binary classifier with ‘easy’ tasks from MD’s original random sampler (positive samples) and ‘difficult’ tasks extracted by our algorithm (negative samples).  This surrogate classifier could then be used to assign a difficulty score to new tasks which could form the basis of a difficulty metric for an experimental setting. Although this falls outside the scope of our manuscript, it is a good idea for future work.

---

> > ### Author Response · Authors · 2022-11-17
> > **Further questions and comments**
> >
> > We thank the reviewer for the time and effort in reviewing our paper and providing constructive suggestions. This has certainly improved our paper. We hope that our response addresses your concerns and comments. Since we are reaching the end of the discussion period, if there are any further questions or comments we can address, please let us know!
> >
> > --Authors

---

> > > ### Comment · Reviewer_trsF · 2022-11-30
> > > **thanks**
> > >
> > > Yes. the authours' rebuttal clarifies my concerns.

---

> > > > ### Author Response · Authors · 2022-12-08
> > > > **Response to Reviewer trsF**
> > > >
> > > > We thank the reviewer for their reply and we are glad that our rebuttal has clarified the concerns!

---

### Official Review · Reviewer_Q1sQ · 2022-10-25

**Confidence:** 4
**Correctness:** 3
**Technical Novelty And Significance:** 2
**Empirical Novelty And Significance:** 3
**Recommendation:** 5

**Clarity, Quality, Novelty And Reproducibility:**


The paper is well written. However, the novelty is somewhat limited, since it can be considered as an extension of the paper[1].

Reproducibility is unclear since there are no codes or links provided.

**Strength And Weaknesses:**

Strength:
1) The paper is writing clearly and easy to read.
2) The paper builds a new test-only few-shot classification benchmark named HARD-META-DATASET++.

Weaknesses:

1) The support samples of "difficult task" is selected by fixing the query samples which means the difficulty of each task is highly correlated with the fixed query data. The hypothesis is somewhat unreasonable, since the FSL aims to adapt to the whole new class not the fixed query samples. The reviewer supposes that the tasks in the HARD-META-DATASET++ may not be a reliable estimator in the test-time in FSL.

2) The proposed method can be seen as an extension of the paper[1], since the paper[1] have already proposed that FSL are extremely sensitive to the data used for adaptation and used a greedy algorithm to find those difficult tasks. Although the proposed method is more efficient, the novelty is somewhat limited.

3)The evaluation process consists 200 tasks using Prototypical Network, which is not enough(suffering from high randomness). In recent literature, the number of evaluation tasks is usually more than 2000.


As for questions, I would like to ask:

1 In FSL, we usually report the mean and the variance of the accuracy over 2000 tasks. The variance of the accuracy also denotes the model's performance of the challenging tasks(The higher variance, the lower accuracy on more difficult tasks). Besides, the average accuracy of several worst cases can also be a good estimator. Why is the accuracy on HARD-MD  a better criterion for evaluation?

2 How to use the "difficult task" in the training phase in FSL? Can the "difficult task" in base classes help the model improve the generality to novel classes?

**Summary Of The Paper:**

The paper studies the failure cases in FSL and proposes an efficient algorithm to extract the difficult tasks from large-scale datasets. Based on the proposed algorithm, the paper builds a new test-only few-shot classification benchmark named HARD-META-DATASET++.

**Summary Of The Review:**

This paper builds a new test-only few-shot classification benchmark named HARD-META-DATASET++. However, I have some concerns that whether the HARD-META-DATASET++ can be a reliable estimation (Please refer to 1) in Weaknesses).

---

> ### Author Response · Authors · 2022-11-13
> **Response to Reviewer Q1sQ**
>
> **Number of Tasks**: We note that widely-used few-shot benchmarks like MetaDataset (MD) and VTAB call for evaluation on 600 tasks per dataset and 1 task per dataset, respectively, rather than 2000 tasks as the reviewer mentioned. Our current version of Hard-MetaDataset++ evaluates on 200 tasks per dataset, however in our revised version, we have increased this to 600 tasks per dataset to match the original MD. This gives HardMetaDataset++ a total of 7200 difficult tasks. We find that the average task accuracy changes only slightly and the relative ordering of the different evaluated methods and subsequent conclusions on Hard-MD++ remains similar. We will release the extracted tasks from Hard-MD++ to the community, so that different methods can be evaluated on a fixed set of difficult tasks. We have added these new results in Appendix (K).
>
>
> **Why accuracy on Hard-MD is a better criterion**:  As mentioned in Sec. (5), we do not propose for Hard-MD++ to replace MD,  but rather for Hard-MD++ to be used as a complementary signal to understand the worst case behavior of few-shot models. Indeed, the mean and variance over task accuracies can provide useful insights into worse-case performance, however, when these tasks are randomly sampled (as it is done in MD) there is no guarantee that these tasks are truly difficult. Our proposed algorithm provides a much more targeted way of obtaining difficult few-shot tasks, and hence Hard-MD++ can be viewed as a much stronger criterion for measuring model robustness to difficult tasks compared to the mean/variance. For example. from Fig. (1), we can observe that the average task accuracy in Hard-MD is lower than the worst-case task accuracy of the random sampler consistently across all the domains. We have added new results in Appendix (O), where we also observe that the tasks in Hard-MD have a fewer number of ways (which can be one source of difficulty) than the filtered out difficult tasks from the random sampler. These results show that the tasks in Hard-MD are harder and have different properties than the difficult tasks from the random sampler.  Furthermore, Hard-MD++ includes 2 datasets with natural image characteristic annotations (e.g. occlusion, blur) and therefore goes beyond current FSL benchmarks in gaining a more nuanced understanding of the robustness of models to various image characteristics.
>
> **Fixed Query Samples / Reliability of Tasks in Hard-MD++**:  Our formulation of a fixed query set and variable support set builds on previous works (Agarwal et.al (2021)) and also enables a more controlled experimental setup for understanding why adaptation to a task may fail (see Sec. (5), where we show that difficult tasks often have support sets with different characteristics to the fixed query sets, thereby capturing a distribution shift). Finally, fixing the query set can also be viewed as similar to fixing a test set, which is a widely-used practice in ML benchmarks.
>
> **Using difficult tasks in the training phase**: We thank the reviewer for this suggestion. For this rebuttal, we re-trained our ProtoNets model with a combination of randomly sampled tasks and difficult talks extracted by our proposed algorithm from the ImageNet split in MD, following the adversarial training procedure proposed in Agarwal et. al(2021). We found that this did not improve the model’s performance on Hard-MD++ (see Appendix (P)), suggesting that there is a need for algorithmic innovation for handling the difficult tasks in Hard-MD++. This observation is similar to Agarwal et. al (2021), who show that adversarial training during meta-training does not help in meta-testing. We have included these analyses in the appendix of our revised manuscript.
>
> **Novelty of Method**:  The primary novelty of our work comes from the fact that we developed a new, efficient algorithm to extract difficult tasks that enabled us to do few-shot failure analysis in a way that was not computationally feasible using existing approaches. More concretely, our algorithm extracts difficult tasks 20x faster than greedy-based methods for fixed-way, fixed-shot tasks and this speedup improves further for realistic variable-way, variable-shot settings where a task can have up to 50 classes and 1000 examples. We would also like to highlight that our paper is not just a simple benchmark paper. Instead we make contributions in three folds: (i) an efficient difficult task extraction algorithm; (ii) a difficult few-shot benchmark based on MD with additional annotated datasets such as ORBIT and CURE. (iii) An extensive empirical study of a wide-range of models on Hard MD++, bringing novel insights in current failure modes.

---

> > ### Author Response · Authors · 2022-11-17
> > **Further questions and comments**
> >
> > We thank the reviewer for the time and effort in reviewing our paper and providing constructive suggestions. This has certainly improved our paper. We hope that our response addresses your concerns and comments. Since we are reaching the end of the discussion period, if there are any further questions or comments we can address, please let us know!
> >
> > --Authors

---

> > > ### Author Response · Authors · 2022-12-08
> > > **Looking Forward to your Response**
> > >
> > > Dear Reviewer Q1sQ,
> > >
> > > We thank you again for your valuable feedback and comments which has helped to strengthen our paper. As the discussion period is ending soon, we would really appreciate if you could let us know if our responses have addressed your concerns. We will be happy to answer any further questions and address any remaining concerns.
> > >
> > > —Thanks
> > >
> > > Authors

---

### Author Response · Authors · 2022-11-13
**Overall and General Comments**

We thank all the reviewers for their comments and suggestions. During the rebuttal period, we have addressed most reviewer comments and are happy to answer any further in the remainder of the discussion period.

The main technical contribution of our paper is a novel optimization-based algorithm for extracting difficult few-shot classification tasks from large-scale vision datasets. The main advantage of our algorithm is that it is at least 20x faster than existing (greedy search-based) approaches. This speedup enables failure mode analyses on large-scale datasets that were previously infeasible with existing approaches. We leveraged this advantage and extracted over 7000 difficult tasks from 12 large-scale vision datasets. We then stress-tested 13 top-performing few-shot classification methods on these tasks, deriving novel insights on the robustness of current methods. To enable the few-shot community to drive future algorithmic innovation in these directions, we collected these tasks into a new test benchmark called Hard-MetaDataset++. We intend for this benchmark to complement current benchmarks (like MD and VTAB) by providing an explicit measure on worst-case performance of few-shot learners. We highlight that while we used our algorithm for wide-scale error analysis and a benchmark, it could flexibly be used for a number of other purposes - including interpretability and debugging of single models (e.g. extracting and inspecting the most difficult tasks from a single domain) and improving general model performance (e.g. filtering out the difficult tasks and only training on the remainder).

Reviewers highlighted that robustness to difficult tasks in few-shot classification is an important problem [R2, R4]  and found our work to be practical, original and our algorithm effective, especially with the large speedup factor [R2, R3]. All reviewers commented that our paper was well-written and easy to understand.

We will release the code and all the tasks from Hard-MD++ upon the decision of ICLR.

Below we list the summary of the major changes based on the reviewers suggestions:
* Update from 200 tasks per dataset to 600 tasks (Appendix (K)) [Q1sQ]
* Ablations with different base models (Appendix (M)) [tAM3]
* New results on understanding difficult tasks with MS-COCO (Appendix (J)) [ trsF, tAM3]
* Ablations on number of optimization steps (Appendix (I)) [yM4m]
* New results on integration of our framework with fine-grained samplers (Appendix (H))[tAM3]
* New results on integration of task2vec with our framework (Appendix (L)) [tAM3]
* New results with transductive approaches (Appendix (Q)) [tAM3]
* New results with adapters (Appendix (R)) [trsF]
* New results on effect of adversarial training with difficult tasks (Appendix (P)) [Q1sQ]
* Refinement of the definition of difficult task in the introduction [tAM3]
* More analysis on the difficulty of using adaptation functions other than ProtoNets (Appendix (N)) [tAM3, yM4m]

---

### Decision · Program_Chairs · 2023-01-20

**Decision:**

Accept: poster

**Justification For Why Not Higher Score:**

As Reviewer taM3 points out, while the submission's contributions are important, they miss out on the opportunity to explore and advance our understanding of what constitutes a hard few-shot learning task.

**Justification For Why Not Lower Score:**

The submission makes important contributions towards designing a tractable, challenging, and relevant few-shot learning evaluation protocol. Its empirical observations are of interest to the community.

**Metareview: Summary, Strengths And Weaknesses:**

The submission looks into the performance of few-shot learners on hard tasks. It defines "hardness" by proxy using a ProtoNets' query set loss and introduces an algorithm for surfacing hard few-shot learning tasks from large-scale vision datasets. Given some query set, the algorithm selects a subset of all possible support examples which maximizes the ProtoNets' query set loss. Based on this, the authors introduce an evaluation protocol called Hard-Meta-Dataset++ that surfaces hard learning episodes from Meta-Dataset data sources and from ORBIT, CURE-OR, and ObjectNet. The paper reports a consistent drop in model performance across all evaluated approaches when compared with the Meta-Dataset evaluation protocol, and it reports similar difficulties on ORBIT, CURE-OR, and ObjectNet (with CLIP's ViT-B being the strongest performer).

Reviewers note the submission's clarity (Q1sQ, tAM3) and the simplicity and effectiveness of its proposed selection algorithm (trsF, yM4m). All reviewers agree on the importance of realisting and challenging few-shot evaluation benchmarks. Concerns over the number of tasks used for evaluation (Q1sQ), the selection algorithm's novelty in light of Agarwal et al. (Q1sQ), missing recent approaches (trsF), and the selection algorithm's reliance on a specific few-shot learner (yM4m, tAM3) have been adequately addressed in the rebuttal.

Reviewer taM3 remains concerned that the definition of "hardness" presented in the submission is unsatisfying, since it does not take into account the intrinsic difficulty of classifying some subset of the test classes, for example. However, they recognize the paper's contributions, especially in light of the rebuttal, and are therefore open to acceptance.

I therefore recommend accepting the submission.

**Note From Pc:**

if the above contains the word "oral" or "spotlight" please see: "oral" presentation means -> notable-top-5% and "spotlight" means -> notable-top-25%. As stated in our emails, we are disassociating presentation type from AC recommendations

**Summary Of Ac-Reviewer Meeting:**

Reviewer taM3 appreciated and was partly convinced by the efforts made by the authors in the rebuttal. They remained concerned that the submission's superficial contribution in terms of advancing the field's understanding of what constitutes a hard few-shot learning task, but conceded that the submission's contributions are of interest to the community and was open to acceptance.

Reviewer Q1sQ noted that the paper is presented clearly and the authors addressed their concerns regarding the number of tasks used for evaluation.

Reviewer trsF did not see any obvious drawbacks with the submission: it tries to analyze and understand hard problems in few-shot learning and does so through comprehensive experiments.